# Pure spin photocurrent in non-centrosymmetric crystals: bulk spin photovoltaic effect

Haowei Xu[1], Hua Wang[1], Jian Zhou [1] & Ju Li [1,2 ✉]

Spin current generators are critical components for spintronics-based information processing. In this work, we theoretically and computationally investigate the bulk spin photovoltaic (BSPV) effect for creating DC spin current under light illumination. The only requirement for BSPV is inversion symmetry breaking, thus it applies to a broad range of materials and can be readily integrated with existing semiconductor technologies. The BSPV effect is a cousin of the bulk photovoltaic (BPV) effect, whereby a DC charge current is generated under light. Thanks to the different selection rules on spin and charge currents, a pure spin current can be realized if the system possesses mirror symmetry or inversion-mirror symmetry. The mechanism of BSPV and the role of the electronic relaxation time $\tau$ are also elucidated. We apply our theory to several distinct materials, including monolayer transition metal dichalcogenides, anti-ferromagnetic bilayer $MnBi_2Te_4$, and the surface of topological crystalline insulator cubic SnTe.

---

[1] Department of Nuclear Science and Engineering, Massachusetts Institute of Technology, Cambridge, MA, USA. [2] Department of Materials Science and Engineering, Massachusetts Institute of Technology, Cambridge, MA, USA. ✉email: liju@mit.edu

Present-day electronics, which utilize the charge degree of freedom of electrons, have revolutionized human civilization. Besides charge, spin is another intrinsic freedom of electrons that can be exploited for information processing. Indeed, spintronics[1,2] is promising for next-generation energy-efficient devices and other novel applications such as quantum computing[3,4] and neuromorphic computation[5]. One of the core challenges[6] of spintronics is the generation of a spin current, and particularly, a *pure* spin current without an accompanying charge current. Until now, there have been a few approaches, such as the interconversion between charge and spin currents by (inverse) spin galvanic effect[7,8] or (inverse) spin Hall effect[9–12], and the interconversion between thermal and spin currents by spin Seebeck effect[13,14] or spin Nernst effect[15,16], etc. These approaches all require electrode contact and patterning, and the response time is usually on the order of nanoseconds or longer. In contrast, optical approaches are noncontact, noninvasive, and can enable ultrafast response time on the order of picoseconds and below.

Several optical approaches for generating spin currents have been proposed; however, these approaches typically require special ingredients, such as the breaking of time-reversal symmetry $\mathcal{T}$ by introducing magnetic elements or circularly polarized light (CPL), and/or special device structures. For example, CPL can selectively couple with spin-up and spin-down states in quantum wells[17], or spin-valley locked systems[18], and the imbalanced population of spin-up and spin-down states could lead to a spin photocurrent. In magnetic materials, it has also been proposed that a linearly polarized light (LPL) can generate a spin current with the shift-current mechanism[19–22]. Alternatively, a spin current can be generated with a mechanism reminiscent of the p–n junction in solar cells[23–25], quantum interference[26,27], or the nonlinear Drude current[28]. Although progress has been made, the generation of spin currents under light is still under-explored. In particular, it is highly desirable to introduce new mechanisms applicable to a broader family of materials.

In this work, we propose a mechanism to generate direct current (DC) spin current based on the nonlinear optical (NLO) theory. This mechanism is a cousin of the bulk photovoltaic (BPV) effect[29,30], whereby DC charge currents can be generated in a uniform crystal under light illumination. The BPV effect, together with other NLO effects, are under intensive research recently, but thus far the attention is mainly on the charge current, while the spin current has long been neglected. Certainly, when the charge flows under light, the spin associated with the carriers are moving as well, which is a spin current. In some situations, the charge current vanishes due to symmetry, but this does not indicate that the carriers are frozen in materials. Indeed, the carriers generally still move under above-bandgap light illumination, which leads to a nonzero pure spin current. A generic picture here is that electrons with opposite (or at least different) spin polarizations travel in the opposite directions so that the net charge current is zero, while the net spin current is nonzero (Fig. 1). We name this effect the bulk spin photovoltaic (BSPV) effect. Here the "voltaic" is defined as $V_{\uparrow\downarrow} \equiv (\mu_\uparrow - \mu_\downarrow)/(-e)$, which is the difference between the chemical potential of spin-up ($\mu_\uparrow$) and spin-down ($\mu_\downarrow$) electrons. This should be compared with the BPV voltage, which may be defined as $U \equiv (\mu_\uparrow + \mu_\downarrow)/(-2e)$. Similar to the BPV voltage $U$, the BSPV voltage $V_{\uparrow\downarrow}$ will not be limited by the bandgap of the material, and the currents will not be limited by the Shockley–Queisser limit.

In the following, we first introduce a unified theory on NLO spin (BSPV) and charge (BPV) currents generation. Then, combining theoretical analysis and ab initio calculations, we elucidate some prominent features of the BSPV. Notably, the only requirements for BSPV are (a) above-direct-bandgap light

illumination, and (b) the breaking of inversion symmetry $\mathcal{P}$, regardless of $\mathcal{T}$. There are no need for any special ingredients such as magnetic materials, special device structures (quantum wells, junctions, etc.), the interference between two pulses, or specific light wavelength or polarization. Hence, BSPV has great convenience in practice and can be readily integrated with existing semiconductor technologies[31,32]. These advantages, together with the flexibilities of optical approaches (dynamic spatial addressability, tunable intensity, wavelength, polarization, etc.), provide a large playground to be explored. These results are useful not only for generating spin currents but also for material characterization and sensing. Many applications that are not envisaged before may become possible. In addition, we also clarify the lattice symmetry requirements for the generation of pure spin current, and the mechanisms (shift- and/or injection-like) for spin current generation under different symmetry conditions and light polarizations.

## Results

**General theory and symmetry analysis.** The NLO charge or spin current under light with frequency $\omega$ can be expressed as

$$J^{a,s^i} = \sum_{\Omega = \pm \omega} \sigma_{bc}^{a,s^i}(0; \Omega, -\Omega)E^b(\Omega)E^c(-\Omega) \quad (1)$$

Here $E(\omega)$ is the Fourier component of the electric field at angular frequency $\omega$. $\sigma_{bc}^{a,s^i}$ is the NLO conductivity, with $a, b, c$ as Cartesian indices. $a$ indicates the direction of the current, while $b$ and $c$ are the polarization of the optical electric field. $s^i$ with $i = x, y, z$ is the spin polarization, while $s^0$ represents charge current. The spin and charge are in the unit of angular momentum $\frac{\hbar}{2}$ and electron charge $e$, respectively. To directly compare the values of the charge and spin current conductivity, we divide the spin current conductivity by a factor of $\frac{\hbar}{2e}$[33]. Equation (1) suggests that the $+\omega$ and $-\omega$ components of the electric field are combined, and a direct current is generated. We derived the formula for $\sigma_{bc}^{a,s^i}$ from quadratic response theory[30,34] (see Supplementary Information). Within the independent particle approximation, the conductivity can be expressed as

$$\sigma_{bc}^{a,s^i}(0; \omega, -\omega)$$
$$= -\frac{e^2}{\hbar^2\omega^2} \int \frac{d\mathbf{k}}{(2\pi)^3} \sum_{mnl} \frac{f_{lm}v_{lm}^b}{\omega_{ml} - \omega + i/\tau} \left( \frac{j_{mn}^{a,s^i}v_{nl}^c}{\omega_{mn} + i/\tau} - \frac{v_{mn}^c j_{nl}^{a,s^i}}{\omega_{nl} + i/\tau} \right) \quad (2)$$

Here the explicit $\mathbf{k}$-dependence of the quantities are omitted. $f_{lm} = f_l - f_m$ and $\omega_{lm} = \omega_l - \omega_m$ are the difference of occupation number and band energy between bands $l$ and $m$. $v_{nl} \equiv \langle n|\hat{v}|l \rangle$ is the velocity matrix element, while $\tau$ is the carrier lifetime, and is set to be 0.2 ps uniformly in this paper. The symmetric real and asymmetric imaginary part of $\sigma_{bc}^{a,s^i}$ correspond to the conductivity under LPL and CPL, respectively. Note that Eq. (2) uses the velocity gauge, while the well-known shift and injection charge current formulae[35] use the length gauge. These two gauges are equivalent[36,37] (Supplementary Information). An advantage of the velocity gauge is that the equations are relatively short and neat, and are easily generalizable to other responses under light, such as valley currents, static magnetization, etc.

The physical mechanism of BSPV can be better understood when compared with BPV. In Eq. (2), $j^{a,s^i}$ with $i \neq 0$ is the spin current operator, defined as[38] $j^{a,s^i} = \frac{1}{2}(v^a s^i + s^i v^a)$. Here $s^i = \frac{\hbar}{2}\sigma^i$ is the spin operator with $\boldsymbol{\sigma}$ as the Pauli matrices. Note that there are lots of debates on the definition of spin current[39–41], see Supplementary Information for detailed discussions. If we define

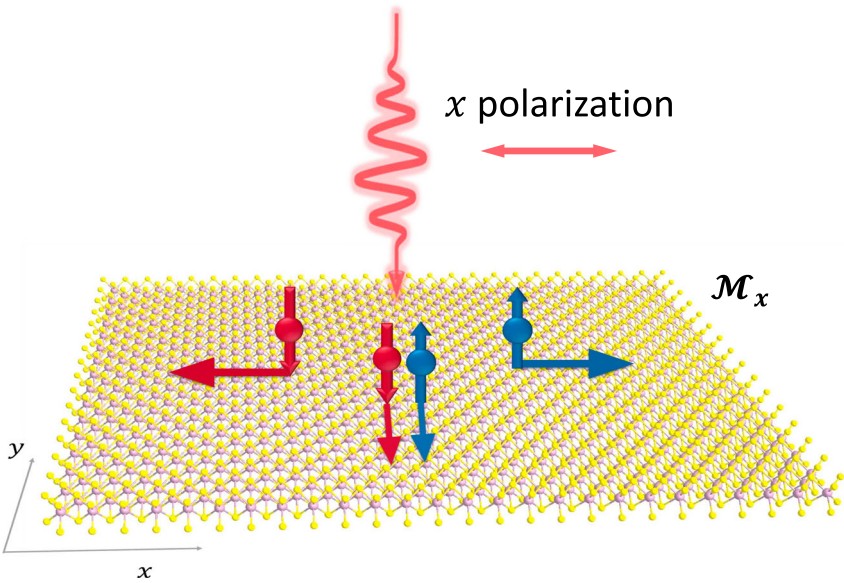

**Fig. 1 A schematic illustration of pure spin and charge current.** The light polarizes in the $x$-direction, while the system has mirror symmetry $\mathcal{M}_x$. In the $x$-direction, spin-up and spin-down states travel in opposite directions, so that the net charge current is vanishing, whereas the net spin current goes to the $+x$-direction. In the $y$-direction, spin-up and spin-down electrons travel in the same direction, leading to nonvanishing charge current but vanishing spin current.

$s^0 = e$, then $j^{a,s^0}$ would indicate the charge current in BPV. The unified formula for spin and charge currents indicates that the DC spin current has a similar physical picture as the BPV, except that spin is a pseudovector; thus, it has different symmetries and selection rules from the charge, which is a scalar. When electron moves, its charge and spin would move simultaneously, leading to the charge and spin current, respectively. However, unlike charge, which is always $-|e|$ for an electron, spin does not necessarily have a specified value. A free electron can have equal probability to have $s_z = \frac{1}{2}$ or $-\frac{1}{2}$. When free electrons move to the right, the spin-$z$ current associated would have an equal probability to be in the right (when $s_z = \frac{1}{2}$) or the left (when $s_z = -\frac{1}{2}$) direction, and the net spin current is thus zero[42]. Therefore, BSPV requires that the electrons have specified spin polarizations (i.e., a spin texture), which can be created by either spin–orbit coupling (SOC), or intrinsic magnetic ordering. Different from the formalism used in ref. [21], Eq. (2) does not require the spin to be a good quantum number or treat spin-up and spin-down states separately, so it can deal with arbitrary spin polarization under SOC. Later, we will show that treating SOC in such a rigorous way is of importance.

Next, we consider symmetry constraints on the conductivity tensor. First, the numerators in Eq. (2) are composed of terms with the format $N_{mnl}^{iabc} = j_{mn}^{a,s^i} v_{nl}^b v_{lm}^c$ ($i \neq 0$) for spin current and $N_{mnl}^{0abc} = v_{mn}^a v_{nl}^b v_{lm}^c$ ($i = 0$) for charge current. Under spatial inversion operation $\mathcal{P}$, one has $\mathcal{P} v_{mn}^a(\boldsymbol{k}) = -v_{mn}^a(-\boldsymbol{k})$, $\mathcal{P} s_{mn}^i(\boldsymbol{k}) = s_{mn}^i(-\boldsymbol{k})$, and $\mathcal{P} j_{mn}^{a,s^i}(\boldsymbol{k}) = -j_{mn}^{a,s^i}(-\boldsymbol{k})$. Thus, $\mathcal{P} N_{mnl}^{iabc}(\boldsymbol{k}) = -N_{mnl}^{iabc}(-\boldsymbol{k})$ for both $i \neq 0$ and $i = 0$. On the other hand, the denominators in Eq. (2) are invariant under $\mathcal{P}$; thus, all components (including charge and spin) of $\sigma_{bc}^{a,s^i}$ should vanish after a summation over $\pm \boldsymbol{k}$ in $\mathcal{P}$-conserved systems. Therefore, the inversion symmetry $\mathcal{P}$ has to be broken for both BPV and BSPV. Regarding time-reversal operation $\mathcal{T}$, one has $\mathcal{T} v_{mn}^a(\boldsymbol{k}) = -v_{mn}^{a*}(-\boldsymbol{k})$ and $\mathcal{T} s_{mn}^i(\boldsymbol{k}) = -s_{mn}^{i*}(-\boldsymbol{k})$ ($i \neq 0$. Here $\bullet^*$ indicates complex conjugate of quantity $\bullet$). For charge current, one has $\mathcal{T} N_{mnl}^{0abc}(\boldsymbol{k}) = -N_{mnl}^{0abc*}(-\boldsymbol{k})$. Thus, the real and imaginary part of

$N_{mnl}^{0abc}$ are odd and even under $\mathcal{T}$, respectively. The imaginary part of $N^{0abc}(\boldsymbol{k})$ contributes to the total charge conductivity after the summation over $\pm \boldsymbol{k}$ in a $\mathcal{T}$-conserved system. Similarly, for spin-$i$ current ($i \neq 0$), one has $\mathcal{T} N_{mnl}^{iabc}(\boldsymbol{k}) = N_{mnl}^{iabc*}(-\boldsymbol{k})$; thus, it is the real part of $N^{iabc}(\boldsymbol{k})$ that contributes to the total spin conductivity. For both charge and spin current, $\mathcal{T}$ does not need to be broken. Generally speaking, spin and charge currents should be generated simultaneously in the absence of $\mathcal{P}$. However, as we will show in detail later, a pure spin current can be realized if the system possesses mirror symmetry $\mathcal{M}^d$, inversion-mirror symmetry $\mathcal{P} \mathcal{M}^d$ or inversion-spin rotation symmetry $\mathcal{P} \mathcal{S}$. The behavior of relevant physical quantities under different symmetry operations is summarized in Table 1.

The carrier lifetime $\tau$ plays a rather important role. Here we use the charge current as the example; a similar analysis applies to the spin current. The DC photocurrent is basically $j^a = \sigma_{bc}^a E^b E^c$. If the system is nonmagnetic, and we use LPL, then it seems that $\mathcal{T}$ should be preserved. In this case, seemingly $\sigma_{bc}^a$ should be zero, because the $j^a$ is odd under $\mathcal{T}$, while $E^b E^c$ is even. However, in practice the nonlinear photocurrent does exist, which is the BPV (shift current). In fact, here $\mathcal{T}$ is effectively broken by dissipation in the thermodynamic second-law sense, characterized by $\tau$. This is related to the well-known paradox regarding microscopic reversibility: if particles in a movie satisfy Newton's equations of motion, then its rewinding version ($t \to -t$) would also; thus, the apparent time-reversal symmetry in the equation of motion. However, if one watches the two movies ($t \to +t$ and $t \to -t$) for long enough time, then the "real" movie is the one with an overall "neater arrangement" of particles at the beginning of play, due to asymmetry in the initial condition. In other words, entropy creation indicates the "arrow of time" and distinguishes between $t$ and $-t$. Therefore, it has been rationalized that the electronic relaxation time $\tau$ is indispensable for the shift current, although the shift-current conductivity $\sigma_{bc}^a$ is (approximately) independent of $\tau$[35].

Dissipation occurs by the scattering of electrons and holes with phonons, etc., which lead to electron–hole recombination. The

**Table 1 The behavior of physical quantities under symmetry operations.**

|  | $v_{mn}^a(k)$ | $s^i(k)$ ($i\neq 0$) | $N_{mnl}^{0abc}(k)$ | $N^{iabc}(k)$ ($i\neq 0$) |
|---|---|---|---|---|
| $\mathcal{P}$ | $-v_{mn}^a(-k)$ | $s_{mn}^i(-k)$ | $-N_{mnl}^{0abc}(-k)$ | $-N_{mnl}^{iabc}(-k)$ |
| $\mathcal{T}$ | $-v_{mn}^{a*}(-k)$ | $-s_{mn}^{i*}(-k)$ | $-N_{mnl}^{0abc*}(-k)$ | $N_{mnl}^{iabc*}(-k)$ |
| $\mathcal{PT}$ | $\widetilde{v}_{mn}^{a*}(k)$ | $-\widetilde{s}_{mn}^{i*}(k)$ | $\widetilde{N}_{mnl}^{0abc*}(k)$ | $-\widetilde{N}_{mnl}^{iabc*}(k)$ |
| $\mathcal{M}^d$ | $(-1)^{\delta_{da}}v_{mn}^a(k')$ | $-(-1)^{\delta_{di}}s_{mn}^i(k')$ | $[d;abc]\times N_{mnl}^{0abc}(k')$ | $-(-1)^{\delta_{di}}[d;abc]\times N_{mnl}^{iabc}(k')$ |
| $\mathcal{PM}^d$ | $-(-1)^{\delta_{ab}}v_{mn}^a(-k')$ | $-(-1)^{\delta_{di}}s_{mn}^i(-k')$ | $-[d;abc]\times N_{mnl}^{0abc}(-k')$ | $(-1)^{\delta_{di}}[d;abc]\times N_{mnl}^{iabc}(-k')$ |

Here $\widetilde{\ }$ indicates $\cdot$ obtained on the $\mathcal{PT}$ partner state, which is degenerate in energy with the original state. $[d;abc]$ is $-1$ and $+1$ if there are odd and even numbers of $d$ within $a, b$, and $c$. For example, $[x; xxx] = -1$, while $[x; xxy] = 1$. $k' = \mathcal{M}^d k$ is the mirror image of $k$ (only the $d$th component of $k$ is flipped).

scattering time $\tau$ is usually on the order of (sub)-picoseconds. In some cases, the spin relaxation time is short, then it can be a source of dissipation as well. Also, in the presence of scattering potentials (from e.g., impurities), there could be skew scattering[43,44] and side jump[45,46], which lead to extrinsic spin/charge currents, as compared with the intrinsic currents studied in this work, that originates from the intrinsic band structure of the perfect crystal. Here we adopt the constant relaxation time approximation and use a constant $\tau = 0.2$ ps for all modes (band index $n$ and wavevector $k$). In reality $\tau$ should be mode dependent (see Supplementary Information for more discussions) of course. This however does not affect the qualitative features of the theory.

To illustrate the theory, we investigate three distinct material systems: (1) monolayer transition metal dichalcogenides (TMD), which are $\mathcal{P}$-broken but $\mathcal{T}$-preserved; (2) antiferromagnetic bilayers $MnBi_2Te_4$ (MBT), which is $\mathcal{P}$- and $\mathcal{T}$-broken but $\mathcal{PT}$-preserving; (3) the {0 0 1} surface of cubic SnTe, which is $\mathcal{P}$-broken, but has double mirror symmetry $\mathcal{M}_x$ and $\mathcal{M}_y$. The results suggest that BSPV is generic and robust in these distinct systems. We only show the NLO charge and spin current under LPL, while the responses under CPL can be found in the Supplementary Information.

**Monolayer TMD.** 2H-phase TMDs are well-studied 2D materials that possess many exotic electronic and optical properties. We take monolayer 2H $MoS_2$ as an example. The atomic structure of monolayer 2H $MoS_2$ (space group $P\bar{6}m2$) is shown in the inset of Fig. 2e, which lacks $\mathcal{P}$, but is invariant under $\mathcal{M}^x$ and $\mathcal{M}^z$. Monolayer TMDs exhibit Zeeman-type (out-of-plane) spin splitting due to the in-plane anisotropy. This could be understood with the effective magnetic field from SOC, expressed as $B_{eff} = \frac{1}{2m_ec^2} p \times \nabla V$, where $m_e$ is the electron mass and $c$ is the speed of light. In monolayer TMDs, the momentum $p$ is in the in-plane ($x$–$y$) direction, while $\nabla V$ is also largely in the $x$–$y$ plane, due to the mirror plane $\mathcal{M}^z$. As a result, $B_{eff}$ is mainly along the out-of-plane direction, leading to the Zeeman-type spin splitting. These arguments are verified by the spin texture $s_{mm}^i(k) = \langle mk|\sigma_i|mk\rangle$ from ab initio calculations. Figure 2a, b show $s_{mm}^z(k)$ for the highest valence band and the lowest conduction band of $MoS_2$, respectively. One can see that $s_{mm}^z(k) \cong \pm 1$ for nearly all $k$-points. Also, $s_{mm}^z(k)$ is opposite near the K and K' valleys, which is the spin-valley locking[47,48].

Here we need to examine constraints on NLO spin or charge current from mirror symmetry $\mathcal{M}^d$ (Table 1). The polar vector $v_{mn}^a$ satisfies $\mathcal{M}^d v_{mn}^a(k) = (-1)^{\delta_{da}} v_{mn}^a(k')$, where $k'$ is the image of $k$ under $\mathcal{M}^d$ (only the $d$-th component flips its sign), whereas the axial vector $s_{mn}^i$ should satisfy $\mathcal{M}^d s_{mn}^i(k) = -(-1)^{\delta_{di}} s_{mn}^i(k')$. Therefore, one has $\mathcal{M}^d N_{mnl}^{0abc}(k) = -N_{mnl}^{0abc}(k')$ when there are odd number of $d$ within $a, b$, and $c$, and the charge current should

vanish in this case. For example, when the system has $\mathcal{M}^x$, then $\sigma_{xx}^{x,s^0}$ and $\sigma_{yy}^{x,s^0}$ should vanish. On the other hand, if $d\neq i$, the spin-$i$ current should vanish when there are even number of $d$ within $a, b$, and $c$, because the $\mathcal{M}^d$ operation on $s^i$ contributes to an additional sign change if $d\neq i$. Therefore, $\sigma_{xx}^{x,s^z}$ and $\sigma_{yy}^{x,s^z}$ could exist in the presence of $\mathcal{M}^x$. Due to the opposite behavior of $N_{mnl}^{0abc}$ and $N_{mnl}^{iabc}$ under $\mathcal{M}^d$, a pure spin current can be generated.

The calculated NLO spin and charge conductivity of monolayer $MoS_2$ under different light polarizations are shown in Fig. 2e, f. One can see that with in-plane polarized light, the nonzero conductivities are complementary for spin and charge currents, consistent with the analysis above. In detail, under the $x$-polarized light, the charge current is along $y$-direction ($\sigma_{xx}^{x,s^0} = 0$ and $\sigma_{xx}^{y,s^0} \neq 0$), whereas the spin-$z$ current is along the $x$-direction ($\sigma_{xx}^{x,s^z} \neq 0$ and $\sigma_{xx}^{y,s^z} = 0$). This indicates that along $x$-direction, equal numbers of spin-up and spin-down electrons are moving oppositely, so the net charge flux is zero, while the net spin flux is nonzero. Along $y$-direction, the spin-up and spin-down carriers move in the same direction, leading to zero spin current but nonzero charge current (Fig. 1). Similar effects occur as well in the case of $y$-polarized light. Interestingly, the spin-$z$ conductivity can be larger than the charge conductivity (in the sense of equivalating $\frac{\hbar}{2} = |e|$). This should be compared with the linear spin Hall effect, where the spin Hall angle (the ratio between the spin conductivity to charge conductivity) is usually on the order of 0.1 and below[49]. We also plot the $k$-specific contribution to the total conductivity, defined as $I_{bc}^{a,s^i}(\omega, k) = \text{Re}\left\{\sum_{mnl} \frac{f_{lm}v_{lm}^b}{E_{ml} - \hbar\omega + i\delta}\left(\frac{j_{mn}^{a,s^i}v_{nl}^c}{E_{mn} + i\delta} - \frac{v_{mn}^c j_{nl}^{a,s^i}}{E_{nl} + i\delta}\right)\right\}$, in Fig. 2c, d for $\sigma_{xx}^{x,s^z}$ and $\sigma_{yy}^{y,s^0}$ at $\omega = 2.8$ eV. The mirror symmetry $k_x \to -k_x$ can be clearly observed.

As discussed before, the generation of spin current requires a spin texture. For $MoS_2$, the spin texture is generated by SOC. When SOC is turned off, the spins of electrons are unpolarized, and the spin current would be zero. This is verified by our ab initio calculations. We artificially rescale the strength of SOC in $MoS_2$ by a factor of $\lambda$, and $\lambda = 0$ ($\lambda = 1$) corresponds to no (full) SOC. The NLO conductivities as a function of $\lambda$ are shown in Fig. 2g, h. One can see that when $\lambda = 0$, the spin conductivity is indeed zero. As $\lambda$ increases, the spins would have more and more specified polarization, and the spin conductivity increase accordingly. In contrast, the charge conductivity is nearly independent of $\lambda$.

**Bilayer antiferromagnetic MBT.** Next, we study the bilayer AFM MBT[50,51], where a large NLO charge current has been reported[52,53]. Each layer of MBT is a septuple layer (SL) in the sequence of Te–Bi–Te–Mn–Te–Bi–Te. The Mn atoms possess

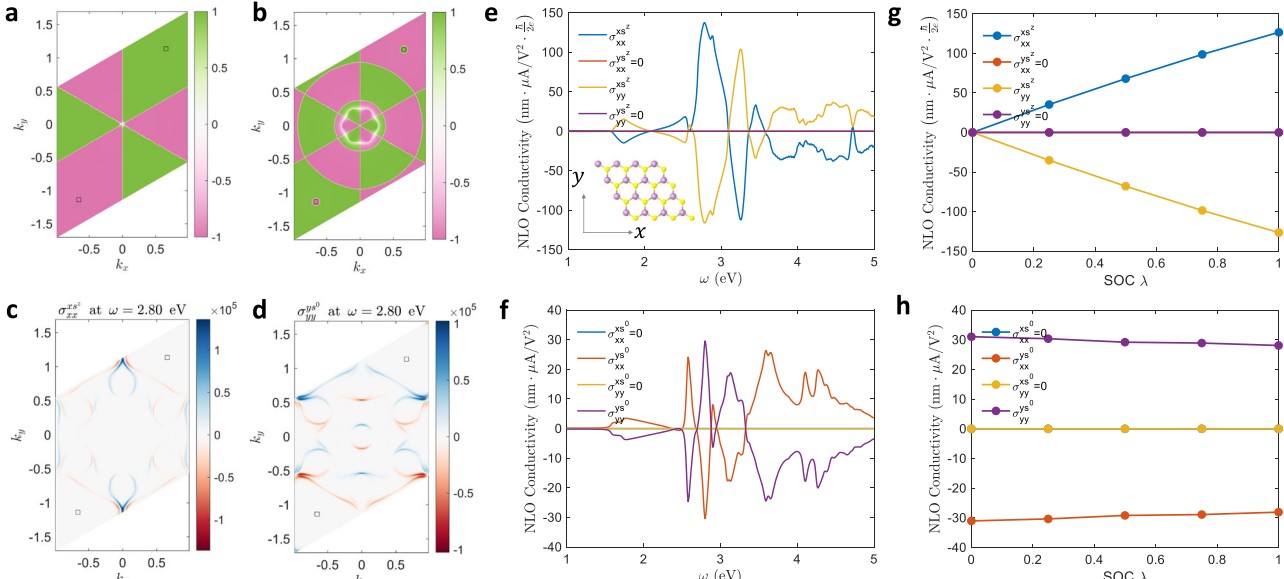

**Fig. 2 NLO spin current of MoS$_2$. a, b** The spin-$z$ texture $s^z_{mm}(\mathbf{k})$ for the **a** highest valence band and **b** lowest conduction band of MoS$_2$. Nearly all $k$-points have $s^z_{mm}(\mathbf{k}) \cong \pm 1$. (**c, d**) $\mathbf{k}$-specified contribution to the total photoconductivity $\sigma^{xs^z}_{xx}$ and $\sigma^{ys^0}_{yy}$. The black boxes in (**a–d**) indicate K and K' points in the BZ. **e, f** The NLO spin-$z$ and charge conductivity. The complementary behavior is clearly observable: the spin and charge currents are in perpendicular directions. Inset of (**e**): the atomic structure of MoS$_2$. **g, h** Peak values of NLO spin (**g**) and charge (**h**) conductivity of MoS$_2$ as a function of SOC strength $\lambda$. The spin conductivity grows linearly with SOC strength, while the charge conductivity is almost independent of SOC strength.

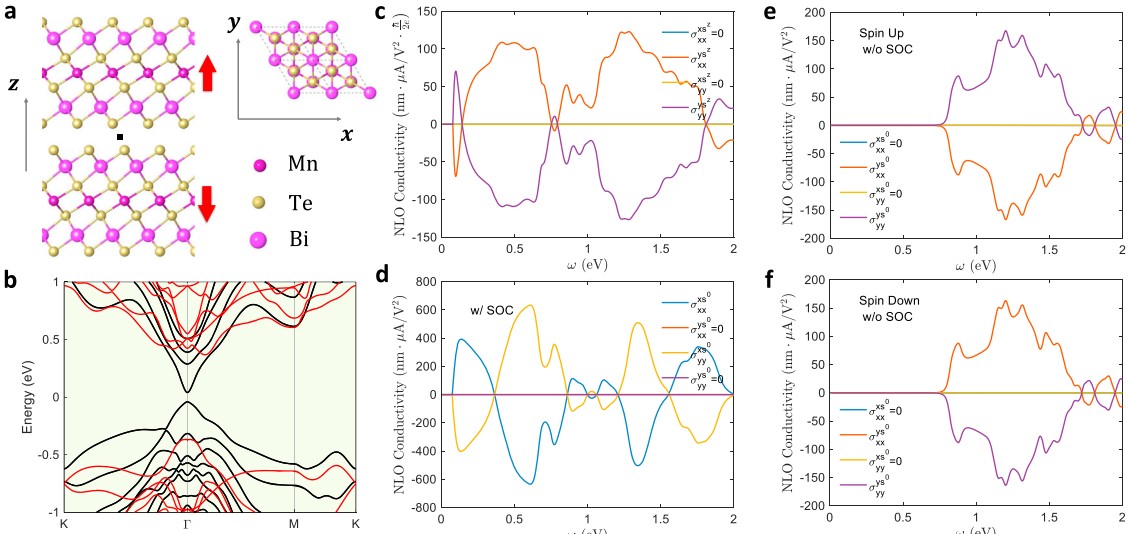

**Fig. 3 NLO spin current of MBT. a** Atomic structure of bilayer MnBi$_2$Te$_4$. The atomic structure has both inversion symmetry $\mathcal{P}$ and mirror symmetry $\mathcal{M}^x$. The inversion center is in between the two layers (black square). The magnetic momentum on Mn is indicated by the red arrows. Considering magnetism, both $\mathcal{P}$ and $\mathcal{M}^x$ break. **b** Band structure of MBT with (black) and without (red) SOC. **c, d** The NLO spin and charge photoconductivity of bilayer MnBi$_2$Te$_4$ with SOC. Both spin and charge currents have nonzero components and exhibit complementary behavior. **e, f** The NLO charge conductivity without SOC. The spin-up and spin-down states are treated separately. The photoconductivity from spin-up and spin-down states are exactly opposite to each other. Therefore, the total charge conductivity is zero. But the spin-$z$ conductivity is nonzero.

magnetic moments $\sim 5\mu_B$, with intra-plane ferromagnetic ordering. Bulk MBT is composed of van der Waals (vdW) stacked SLs with inter-plane AFM ordering, and the AFM nature persists when MBT is thinned down to multiple atomic layers. In particular, bilayer MBT is a compensated AFM insulator, whose atomic structure is shown in Fig. 3a. The ground state magnetic moments are pointing along the $z$-direction with a magnetic point group of $\bar{3}'m'$. The atomic

structure of bilayer MBT is invariant under $\mathcal{P}$ and the inversion center lies in the vdW gap between the two layers (black square in Fig. 3a). However, when one considers magnetism, both $\mathcal{P}$ and $\mathcal{T}$ are broken. Nevertheless, AFM bilayer MBT is invariant under the combined operation $\mathcal{PT}$. Similarly, we find that $\mathcal{PM}^x$ is also preserved. According to the previous analysis (Table 1), we know that $\mathcal{PM}^d v^a_{mn}(\mathbf{k}) = -(-1)^{\delta_{ab}} v^a_{mn}(-\mathbf{k}')$ and $\mathcal{PM}^d s^i_{mn}(\mathbf{k}) = -(-1)^{\delta_{di}} s^i_{mn}(-\mathbf{k}')$. Then, one can see that when

$d \neq i$, $N^{0abc}$ ($N^{iabc}$) should vanish after Brillouin zone integration when there are even (odd) number of $d$ within $a$, $b$, and $c$. Therefore, one can still obtain a pure spin current in systems with $\mathcal{PM}^d$ due to the different selection rule on charge and spin currents.

The band structures of bilayer MBT with and without SOC are shown in Fig. 3b. The bandgap is ~0.1 eV and is located at the Γ point when the SOC effect is included, whereas it is ~0.7 eV and is indirect without SOC. As shown in Fig. 3c–f, the SOC also makes a significant difference in the NLO spin and charge conductivity. When SOC is turned off, $s^z$ is a good quantum number. States with $s^z = \pm 1$ are strictly degenerate in an AFM system and can be treated separately. The NLO conductivities without SOC are shown in Fig. 3e, f, where one can see that the charge current from spin-up ($j_\uparrow$) and spin-down ($j_\downarrow$) states are exactly opposite to each other. Consequently, the total charge current $j^{s^0} = j_\uparrow + j_\downarrow$ is zero. However, the total spin-z current $j^{s^z} = j_\uparrow - j_\downarrow$ is nonzero. Therefore, a pure spin current without any charge current is predicted, which comes from the inversion-spin rotation symmetry $\mathcal{PS}$. These results are consistent with those in ref. [19], where several other well-known AFM materials such as NiO and BiFeO$_3$ were studied.

However, SOC would break $\mathcal{PS}$, and thus lead to a nonzero charge current. Due to the $\mathcal{PM}^x$ symmetry, the charge current is perpendicular to the spin-z current (Fig. 3c, d). We also artificially rescale the strength of SOC by a factor of $\lambda$, as done in the MoS$_2$ section (see Supplementary Information). It is found that the charge conductivity increases with $\lambda$. This is because with a larger $\lambda$, the spin and orbital degrees of freedom are coupled more strongly, and inversion-spin rotation symmetry $\mathcal{PS}$ is broken to a greater extent; thus, the charge conductivity would be larger. These results suggest that while SOC enables spin current in nonmagnetic materials such as MoS$_2$, it would adversely hinder the generation of pure spin current in some cases. Also, SOC should be treated rigorously when studying both the spin current and the charge current.

**2D surface of 3D topological materials**. Topological insulators[54–56] (TIs) and topological semimetals[57,58] have attracted intense interest in recent years. In TIs, the bulk states are insulating with a finite bandgap, while the surface states are (semi)-metallic with symmetry-protected vanishing bandgap, which has potential applications in electronic and spintronic devices. One salient feature of the surface states is the spin-momentum locking, which could prevent the electrons from backscattering and facilitate spin manipulations[59–61]. In addition, the inversion symmetry $\mathcal{P}$ is naturally broken on the surfaces, even if the bulk possesses $\mathcal{P}$. Therefore, the NLO charge[62] and spin current can be induced solely on surfaces, while the bulk remains silent.

Here we take the topological crystalline insulator (TCI)[63,64] cubic SnTe as an example. The bulk SnTe has space group $Fm\bar{3}m$, and is inversion symmetric inside the 3D crystal, which forbids BPV/BSPV in the bulk interior. But the 2D surfaces of this 3D crystal would lose the inversion symmetry, and therefore can support both BPV and BSPV. Here we consider the {0 0 1} surface, which has a four-fold rotational symmetry and double mirror symmetries $\mathcal{M}^x$ and $\mathcal{M}^y$ (Fig. 4a). The spectrum function $A(\boldsymbol{k}, \omega)$ of the {0 0 1} surface is obtained with iterative Green's function method[65,66] and is shown in Fig. 4b, c. In Fig. 4b, $A(\boldsymbol{k}, \omega)$ along high-symmetry lines in the BZ is presented, and the gapless surface states can be clearly observed. In Fig. 4c, $A(\boldsymbol{k}, \omega)$ near X̄ point in the BZ with selected energy $\omega = -0.2$, 0, and 0.2 eV are plotted. One can see that $A(\boldsymbol{k}, \omega)$ can have significant

values on the same $\boldsymbol{k}$-point with different $\omega$, enabling strong interband transitions and significant photocurrents. In addition, the surface spin textures are plotted as black arrows. The nonzero $s^x$ and $s^y$ components indicate that one can obtain spin-$x$ and spin-$y$ currents.

According to our previous symmetry analysis, under in-plane polarized light ($b$, $c = x$ or $y$), no NLO charge or spin-z current can be generated on the {0 0 1} surface, due to the double mirror symmetry $\mathcal{M}_x$ and $\mathcal{M}_y$. However, it is possible to have nonzero spin-$x$ and spin-$y$ currents, which is verified by our ab initio calculations. We use a slab model to compute the surface NLO spin and charge conductivity. To distinguish the contribution from only one surface of the slab, we define a projection operator[67] $P_l = \sum_{i \in l} |\psi_i\rangle\langle\psi_i|$. Here $|\psi_i\rangle$ are atomic orbitals centered on the $l$-th atomic layer. Then, we replace the current operator $j$ in Eq. (2) with $P_l j P_l$, and the resultant conductivity can be layer distinguished (on the $l$th layer). Note that there could be nonzero cross-terms $P_l j P_m$ (with $l \neq m$), indicating the interference between the $l$th and $m$th layer. From our computations, even for neighboring layers with $m = l \pm 1$, the contribution from $P_l j P_m$ is quite small (<10%). Here for a conceptual demonstration of our theory, we only consider $P_{l=1} j P_{l=1}$ and calculate the contribution from the out-most layer. NLO spin-$x$ and spin-$y$ conductivities are plotted in Fig. 4d. One can see that the maximum value of $\sigma_{yy}^{ys^x}$ can reach 500 nm × μV/A$^2$ × $\frac{\hbar}{2e}$. We would like to emphasize again that under the light field with in-plane polarization, NLO charge current is absent on this {0 0 1} surface; therefore, a pure spin current without any charge current can be generated due to the double mirror symmetries. Such methodology can also be used to distinguish surface and bulk states and to probe the surface states. There may be other systems that possess double mirror symmetries as well, such as monolayer FeSe[68], which may be good candidates for pure spin current generation.

## Discussions

Before concluding, we would like to make some remarks. First, it is well known that BPV has potentially shift and injection current contributions. The shift mechanism comes from the fact that the wavefunction center of the electron and hole band states are different, leading to an electric dipole upon photon absorption. On the other hand, the injection mechanism comes from the fact that the electron and holes have different velocities, and that the coherent $\boldsymbol{k}$ and $-\boldsymbol{k}$ excitations are imbalanced, leading to $\boldsymbol{k}$ and $-\boldsymbol{k}$ asymmetry in steady-state population and a net current. These facts are more evident if we transform Eq. (2) into the length gauge, as shown in Supplementary Information . In a $\mathcal{T}$-conserved system, the DC charge currents under LPL and CPL have shift and injection mechanism, respectively[35]. In contrast, for the DC spin current, the mechanism under LPL and CPL should be injection-like and (shift + injection)-like (see Supplementary information). Here the shift- (injection-) current is defined by the conductivity scaling relationship as $\propto \tau^0$ ($\tau^1$). Therefore, the spin conductivity in Figs. 2e and 4d can be further enhanced if a larger $\tau$ is used (see Supplementary Information). The different mechanisms for spin and charge current come from the different behavior of $N_{mnl}^{iabc}(i \neq 0)$ and $N_{mnl}^{0abc}$ under $\mathcal{T}$-operation. Note that in $\mathcal{T}$-conserved systems, the shift spin current should vanish under LPL, consistent with the arguments in ref. [20] We have done similar analyses on mechanisms of current generation under different symmetry conditions, and the results are listed in Table 2. These results are also computationally verified by varying $\tau$ (see details in Supplementary Information).

Second, as shown above, a pure spin current induced by mirror symmetry is usually accompanied by a charge current in the transverse direction (except for the {1 0 0} surface states of cubic

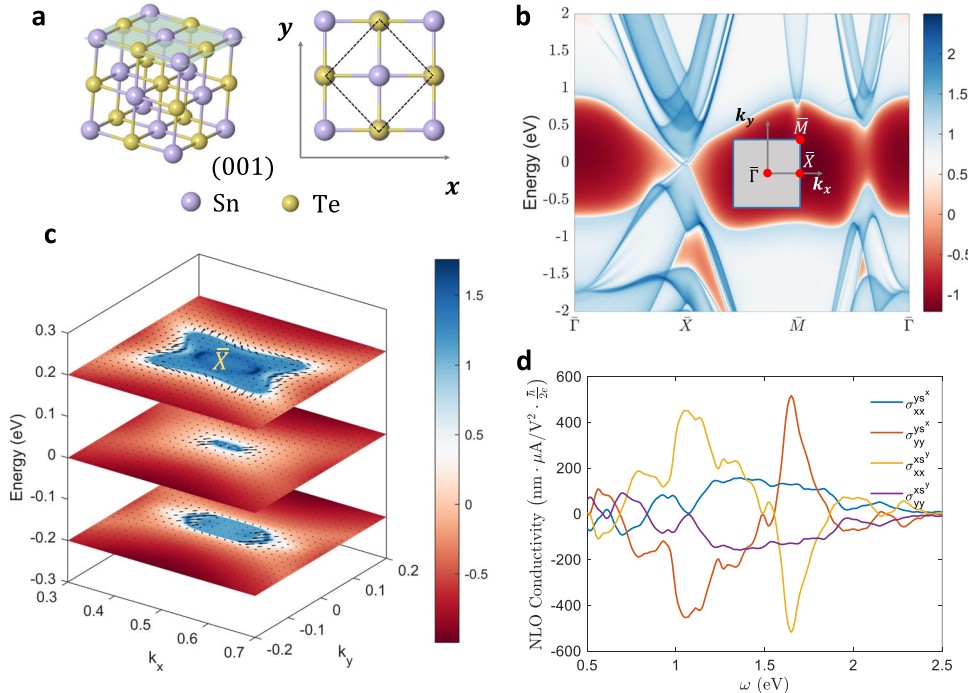

**Fig. 4 NLO spin current on the (0 0 1) surface of SnTe. a** The atomic structure of SnTe. In the left panel the {0 0 1} face is painted in light green, which possesses double mirror symmetries $\mathcal{M}^x$ and $\mathcal{M}^y$. The dashed box in the right panel indicates the primitive cell on the surface. **b** The surface spectrum function $A(\boldsymbol{k}, \omega)$ on high-symmetry lines in the BZ. The gapless surface states can be observed. **c** The surface spectrum function $A(\boldsymbol{k}, \omega)$ over the BZ for selected $\omega = -0.2$, 0, and 0.2 eV. $k_x$ and $k_y$ are in the unit of reciprocal lattices. The surface spin textures are indicated by the black arrows. Color scheme (red to blue) in (**b**, **c**) represents surface state contribution. The color bars are in logarithmic scale, and the energy is offset to the valence band maximum. **d** The NLO spin current conductivity with $x$ and $y$ spin polarizations. Note that all charge and spin-$z$ current components are vanishing due to $\mathcal{M}^x$ and $\mathcal{M}^y$.

**Table 2 Mechanisms for NLO charge and spin current generation under different material symmetries and light polarizations.**

| | $\mathcal{P}$-conserved | $\mathcal{P}$-broken, $\mathcal{T}$-conserved | $\mathcal{P}$-broken, $\mathcal{T}$-broken $\mathcal{PT}$-conserved | $\mathcal{P}$-broken, $\mathcal{T}$-broken $\mathcal{PT}$-broken |
|---|---|---|---|---|
| DC charge current (BPV) | No | LPL $\Longleftrightarrow$ shift<br>CPL $\Longleftrightarrow$ injection | LPL $\Longleftrightarrow$ injection<br>CPL $\Longleftrightarrow$ shift + injection | LPL $\Longleftrightarrow$ shift + injection<br>CPL $\Longleftrightarrow$ shift + injection |
| DC spin current (BSPV) | No | LPL $\Longleftrightarrow$ injection<br>CPL $\Longleftrightarrow$ shift + injection | LPL $\Longleftrightarrow$ shift<br>CPL $\Longleftrightarrow$ injection | LPL $\Longleftrightarrow$ shift + injection<br>CPL $\Longleftrightarrow$ shift + injection |

For the shift mechanism, the conductivity contribution is independent of the carrier lifetime $\tau$. For the injection mechanism, the conductivity contribution scales linearly with $\tau$.

SnSe, with double mirror symmetry $\mathcal{M}^x$ and $\mathcal{M}^y$). It is possible to achieve a pure spin current without any charge current at all, if the system possesses inversion-spin rotation symmetry $\mathcal{PS}$. One can see that $\mathcal{PS}N_{mnl}^{0abc}(\boldsymbol{k}) = -\hat{N}_{mnl}^{0abc}(-\boldsymbol{k})$, where $\hat{}$ indicates $\cdot$ obtained on the spin partner state. Therefore, the charge current should identically be zero in the presence of $\mathcal{PS}$. On the other hand, $\mathcal{PS}N_{mnl}^{iabc}(\boldsymbol{k}) = -e^{i\phi}\hat{N}_{mnl}(-\boldsymbol{k})$, where $e^{i\phi}$ is a phase factor induced by the spin rotation operation on $s^i$. Thus, the spin current does not have to vanish. In fact, $\mathcal{PS}_z$, where $\mathcal{S}_z$ flips the spin-up and spin-down states, is the origin of the vanishing charge current of MBT when SOC is ignored. In practice, a skyrmion lattice, or magnetic materials with canted or all-in-all-out magnetic ordering, can be an ideal platform for the generation of pure spin current without any charge current.

Third, the NLO conductivity in Eq. (2) is obtained from the quadratic response theory. It essentially is $\mathrm{Tr}\left(j^{(0)}\rho^{(2)}\right)$, where $j^{(0)}$ is the current operator independent of the electric field $E$, while $\rho^{(2)}$ is the second-order perturbation in the density matrix and is

proportional to $E^2$. There could be other mechanisms for the generation of spin/charge current. For example, there could be an anomalous velocity, which leads to an additional term $j^{(1)}$ in the current operator that is linearly dependent on $E$. $j^{(1)}$ can contribute to an NLO conductivity from $\mathrm{Tr}\left(j^{(1)}\rho^{(1)}\right)$, where $\rho^{(1)}$ is the first-order perturbation in the density matrix. Note that this contribution should vanish for all the material systems studied in this work.

Finally, we would like to briefly discuss how the spin current can be detected. There are well-established approaches for detecting the spin current generated by, e.g., spin Hall effect[9]. For example, with an open-circuit setup (Supplementary Information and Figure S2a), the spin would accumulate on the ends of the source material. The spin accumulation can be measured by magneto-optic effects such as Kerr rotation or Faraday effect[69]. Also, in a close circuit setup (Supplementary Information and Figure S2b), the spin current source is sandwiched between two metallic leads (e.g., Pt). The light-induced spin current is

transmitted to the metallic leads. An inverse spin Hall voltage would be generated transverse to the spin current[70–72], and the spin current can be measured by the inverse spin Hall voltage. Assuming a spin conductivity of 100 μA/V$^2$ $\frac{\hbar}{2e}$, an electric field as small as 100 V/m would generate a spin current density of 1 A/m$^2$ $\frac{\hbar}{2e}$. Assume a spin Hall angle of 0.1, the current density in the metallic lead would be 10 A/m$^2$, which can be detectable.

In conclusion, we demonstrate a generic picture of spin photocurrent generation with nonlinear light–matter interactions. By symmetry analysis, we reveal that this effect does not have any special requirements, except for the inversion symmetry breaking. Thus, it applies to a wide range of materials and extended defects like surfaces, stacking faults, grain boundaries, and dislocations. If the system possesses mirror symmetry or inversion-mirror symmetry, a pure spin current can be realized. Our theory is verified with ab initio calculations in several material systems, and the spin current conductivity is found to be comparable or even bigger than its charge BPV cousin. The predicted BSPV platforms can be readily integrated with existing semiconductor technologies. They may find applications in next-generation ultrafast spintronics and quantum information processing.

## Methods

The first-principles calculations are based on density functional theory (DFT)[73,74] as implemented in the Vienna ab initio simulation package[75,76]. The exchange–correlation interactions are treated by a generalized gradient approximation in the form of Perdew–Burke–Ernzerhof[77]. Core and valence electrons are treated by projector augmented wave method[78] and plane-wave basis functions, respectively. For DFT calculations, the first Brillouin zone is sampled by a Γ-centered $k$-mesh with grid density of at least $2\pi \times 0.02 \text{A}^{-1}$ along each dimension. The DFT + U method is adopted to treat the $d$ orbitals of Mn atoms in MBT ($U = 4.0$ eV). Then a tight-binding (TB) Hamiltonian is constructed from DFT results with the help of the Wannier90 package[79]. The TB Hamiltonian is utilized to calculate the NLO charge and spin conductivity according to Eq. (2) on a finer $k$-mesh. The $k$-mesh convergence for BZ integration is well tested. In practice, the BZ integration in Eq. (2) is carried out by $k$-mesh sampling with $\int \frac{dk}{(2\pi)^3} = \frac{1}{V} \sum_k w_k$, where $V$ is the volume of the simulation cell in real space and $w_k$ is weight factor. However, for 2D materials, the definition of volume $V$ is ambiguous, because the thickness of 2D materials is ambiguously defined[80]. Thus, we replace volume $V$ with the area $S$, and the 2D and 3D conductivity satisfy $\sigma_{2D} = L\sigma_{3D}$, where $L$ is an effective thickness of the material.

## Data availability

The authors declare that the main data supporting the findings of this study are available within the article and its Supplementary information files.

## Code availability

The MATLAB code for computing the NLO conductivities is available from the corresponding author upon reasonable request.

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

## Acknowledgements

This work was supported by an Office of Naval Research MURI through grant #N00014-17-1-2661. We are grateful for the insightful suggestions by Dr. Zhurun Ji.

## Author contributions

H.X. and J.L. conceived the idea and designed the project. H.X. derived the theories. H.X. performed the calculations and wrote the paper with the help of H.W. and J.Z. J.L. supervised the project. All authors analyzed the data and contributed to the discussions of the results.

## Competing interests

The authors declare no competing interests.
