## [Peer Review File · Nature Communications]

Reviewer #1:

Remarks to the Author:

Dear Editors,

The manuscript titled "Pure spin photocurrent in non-centrosymmetric crystals: bulk spin photovoltaic effect" shows that "nonlinear optical (NLO) effect can be used to generate pure spin currents" "if the system possesses additional mirror symmetry or inversion-mirror symmetry". Using a computational approach based on second order response, the authors compute a pure spin current for a few example materials to show the generation of a photo-spin current that significantly exceeds the photo charge current response.

The authors are motivated by "One of the core challenges of spintronics is the generation of the spin current, and particularly, a pure spin current without an accompanying charge current", which motivated by Ref 6, they believe "find applications in next generation energy efficient and ultrafast spintronics". Despite the somewhat extensive discussion, because of the rather extensive work in this field, I am not entirely convinced that the motivation for the work reaches the level of impact required for consideration in Nature Communications. Below I elaborate the reasons for my concern.

While it hasn't been called "injection current", the idea of generating spin currents through photoexcitation of semiconductors such as GaAs has been around for a while. This is clear from the fact that Fig 1a of PRL 96, 246601 (2006) has some at least superficial resemblance to Fig 1 of the current manuscript though the role of mirror symmetry is not as emphasized. As I understand it from the manuscript, the mirror symmetry leads to "a pure spin current can be realized, which is highly desirable for energy-efficient spintronics." This confuses me because as pointed out in Ref 1, a pure spin current does not guarantee lack of dissipation. Specifically, in this case, the optical excitation generates electron-hole pairs, which are highly energetically excited. I expect the relaxation of this excitation to lead to significant dissipation at some point. It seems that the authors are pointing out that TCIs such as SnSe, SnTe have an advantage relative to the GaAs example because of the double mirror symmetry, which should be emphasized more.

A more serious omission in terms of references in my view is PHYSICAL REVIEW B 102, 081402(R) (2020), which has proposed an injection-based pure spin photocurrent in the quantum spin-Hall edge states of TMDs. The mechanism is quite similar to the one in the present work, though this is for a one dimensional edge in a two dimensional system.

Apart from motivational issues that I raised above, I have a few suggestions for the presentation of the formalism. The symmetry argument presented between lines 125-135, which is one of the main points of the manuscript seems to be a bit sloppy. This is because while Eq. 2, whose numerator is being discussed depends on off-diagonal matrix elements of the spin and velocity operators, they appear to be treated here as numbers. Also it would be good to clarify this.

Another technical aspect is the treatment/interpretation of the scattering time τ , which plays a somewhat important role. In the shift mechanism the time τ is a dephasing or scattering time for the electron-hole pair rather than the equilibration time suggested here. This has been recently clarified in PHYSICAL REVIEW B 101, 045201 (2020), where the second order response formula for the charge BVPE is shown to be equivalent to the dipole moment of electron hole pairs if τ is interpreted as a scattering time. In this sense, the injection mechanism is rather physically different and generates a current from a difference of velocities of the different excited charge carriers. This current survives for as long as the electrons/holes maintain their direction, leading to a different interpretation

for τ . I think these subtleties are lost in the second order response formula. Also the electron-hole pair interpretation provides a direct understanding of the symmetries in table 2. Shift currents occur when the excited pair has a dipole moment. Injection currents on the other hand are a result of unequal excitation of carriers of different velocities.

In summary, while I think the work is technically interesting, I am not sure I am convinced that the authors have clarified the motivation for this work that would lead to a high level of impact suggested for publication in Nature Communications.

Reviewer #2:

Remarks to the Author:

This is a timely work which shows theoretically how the nonlinear optical effect could be used to generate pure spin photocurrent in non-centrosymmetric crystals. These findings are illustrated on examples from three classes of materials, transition metal dichalcogenides (MoS₂), antiferromagnetic topological insulator (MnBi₂Te₄), and the surface state of the topological crystalline insulator (SnTe). Both the topic of generating (electrically, or optically) spin current and the considered materials classes are actively studied, so the findings of this manuscript have potential many implications. Specifically, as noted, for example, in Ref. 1, spin current provides an important element for spintronic applications as well as a tool to probe materials properties. Methods which would allow for a robust generation of pure spin current therefore have broad ramifications. The authors complement well their symmetry analysis and conductivity calculations by the materials-specific ab-initio calculations.

To provide a better context of this work and further explain its relevance, it would be helpful to give several explanations and clarify its underlying assumptions.

1) Adding a spin degree of freedom can generalize various photovoltaic effects that have been studied in common semiconductors and their junctions. See, for example, Appl. Phys. Lett. 79, 1558 (2001), Phys. Rev. Lett. 88, 066603 (2002), or related experiments, Nat. Commun. 4, 2068 (2013), which may be helpful to mention. Since the spin current is not conserved it may also be important to identify the position at which it is evaluated. Even in a simple p-n junction, illuminated by circularly-polarized light, a different voltage dependence of the spin and charge currents allows for a generation of pure spin current.

2) This non-conservation of the spin current has led to lots of debates about its definition, when the prediction of the spin Hall effect from Refs. 10, 11, was revisited 30 years later. It would help to clarify how the employed definition for spin current compares to the one from Phys. Rev. Lett. 96, 076604 (2006), which also recalls a cautionary work of Rashba [Phys. Rev. B 68, 241315(R) (2013)].

3) What are the employed assumptions for loss mechanisms of pure spin photocurrents? For example, in a simple situation analyzed in 1) in addition to the spin relaxation, the spin current can be lost through carrier recombination. This spin dynamics and spin-orbit coupling for electrons and holes are inequivalent.

4) Perhaps some clarification for the points above could be given by moving the p.15 comment about focusing only on intrinsic currents to an earlier place in the manuscript. Even in such intrinsic regime, how can one understand the corresponding time/length scales for the presence of spin photocurrents?

5) For an experimental detection of spin photocurrent, could one use spin extraction [predicted in Phys. Rev. Lett. and measured in Nat. Commun. noted above]?

6) References are not given in a uniform style. Some journal names are abbreviated, some not. Ref. 10 has typos and repeated authors, extra number appear in Ref. 27, 45, 67. Are Refs. 31, 40, 47 complete?

This manuscript describes an important opportunity to realize spin photocurrent in a wide range of materials. Following these clarifications, it will be easier to assess its suitability for a broad readership and a publication in Nature Communications.

Reply to Reviewers

Reviewer #1:

The manuscript titled "Pure spin photocurrent in non-centrosymmetric crystals: bulk spin photovoltaic effect" shows that "nonlinear optical (NLO) effect can be used to generate pure spin currents" "if the system possesses additional mirror symmetry or inversion-mirror symmetry". Using a computational approach based on second order response, the authors compute a pure spin current for a few example materials to show the generation of a photo-spin current that significantly exceeds the photo charge current response. The authors are motivated by "One of the core challenges of spintronics is the generation of the spin current, and particularly, a pure spin current without an accompanying charge current", which motivated by Ref 6, they believe "find applications in next generation energy efficient and ultrafast spintronics". Despite the somewhat extensive discussion, because of the rather extensive work in this field, I am not entirely convinced that the motivation for the work reaches the level of impact required for consideration in Nature Communications. Below I elaborate the reasons for my concern.

Reply: We thank the reviewer for these comments. Here we briefly list the motivations and novelties of our work:

- 1) We derived a unified theory of the generation of spin current under light illumination. Nonlinear optical (NLO) approaches have attracted great interest recently, as they can be non-contact (without e.g., electrochemical electrode deposition), non-destructive (does not induce unwanted impurities), and ultrafast (timescale on the order of picoseconds or even femtoseconds). Most previous theoretical and experimental efforts focus on how second-harmonics waves, charge current, etc. can be generated under light in different materials, while our theory points out the possibility of spin current generation with NLO effects. The bulk spin photovoltaic effect is the lowest-order NLO effect that can be used to generate dc spin current.
- 2) Our bulk spin photovoltaic effect is a universal and robust mechanism for spin current generation. Particularly, the only requirement to generate spin photocurrent is inversion symmetry breaking. There is no need for any special ingredients such as magnetic materials,

special device structures (quantum wells, junctions, etc.), the interference between two pulses, or specific light wavelength or polarization, which are required in many previous works. This would provide great flexibility in practice. Also, our theory applies to semiconductors, thus it may be readily integrated with existing semiconductor technologies, which is of significant importance as outlined by Albert Fert [*Rev. Mod. Phys.* **80**, 1517 (2008)]. These advantages, together with the flexibilities of optical approaches (dynamic spatial addressability, tunable intensity, wavelength, polarization, etc.), provide a large playground to be explored. Many spintronic applications (e.g., spin injection into semiconductors, spin FET, spin battery, and spin-dependent energy-harvesting [*Commun. Mater.* **1**, 24 (2020)]) may be realized in an easier fashion with such an optical approach. For example, spin current can be generated in bulk GaAs under sunshine, as we do not need circularly polarized light.

- 3) We explicitly reveal that the symmetry condition for hosting *pure* spin current is to possess mirror symmetry \mathcal{M} , inversion-mirror symmetry \mathcal{PM} , or inversion-spin rotation symmetry \mathcal{PS} . This provides a simple but effective guideline for pure spin current generation under light. Particularly, we studied the surface states of a topological material SnTe, whose double mirror symmetry indicates that the charge current all vanishes, while the spin current can survive on the surfaces. These results are useful not only for generating pure spin currents, but also for the probe of surface states properties of topological materials.
- 4) We also made several other theoretical advances. For example, we clarify and compare the mechanism (shift and/or injection mechanisms) for spin current generation under different symmetry conditions (\mathcal{T} , \mathcal{PT} , etc.) and different light polarizations (linear and circularly polarized light). This illuminates the microscopic mechanisms for spin current generation. We also clarified the important role of the spin-orbit coupling (SOC). On one hand, it can lead to a spin texture in non-magnetic materials, which is necessary for spin current generation. But on the other hand, in many cases SOC breaks the inversion-spin rotation symmetry \mathcal{PS} , hindering the realization of pure spin current.
- 5) Besides the above advances on the physical mechanism for spin current generation, we also computationally studied the bulk spin photovoltaic effect with several different material classes that are attracting great interest recently, which may guide the potential applications for these materials. It also helps the development of next-generation devices with next-generation materials.

We have revised several parts of our manuscript to strengthen the motivation and novelties of our work, including,

- On page 4, we added, “Here we would like to emphasize that the only general requirement for our NLO spin current is \mathcal{P} breaking, and there is no need for any special ingredients such as magnetic materials, special device structures (quantum wells, junctions, etc.), the interference between two pulses, or specific light wavelength or polarizations. This would provide great convenience in practice and can be readily integrated with existing semiconductor technologies. These flexibilities, together with the flexibilities of optical approaches (dynamic spatial addressability, tunable intensity, wavelength, polarization, etc.), provide a large playground to be explored. Many applications that are not envisaged before may become possible.”
- On page 4, we added, “Particularly, when double mirror symmetries, or inversion-spin rotation symmetry \mathcal{PS} are present, the charge current would all vanish (even in the transverse directions), while the spin current can survive. These results are useful not only for generating pure spin currents, but also for material characterization.”
- On page 4, we added, “We also clarify the mechanisms (shift and/or injection like) for spin current generation under different symmetry conditions (\mathcal{P} and \mathcal{T}) and under light with different polarization (LPL and CPL).”
- On page 4, we added, “....., where the “voltaic” is defined as $V_{\uparrow\downarrow} \equiv \frac{\mu_{\uparrow} - \mu_{\downarrow}}{-e}$, the difference in chemical potential (μ) between spin-up (\uparrow) and spin-down (\downarrow) electrons, unlike the BPVE voltage that may be defined as $U \equiv \frac{\mu_{\uparrow} + \mu_{\downarrow}}{-2e}$. Similar to the BPVE voltage U , $V_{\uparrow\downarrow}$ will not be limited by the bandgap of the material, and the currents will not be limited by the Shockley–Queisser limit.”
- On page 8, we added, “..... The results suggest that our mechanism for spin current generation is general and robust in these distinct systems.”
- On page 13, we added, “These results suggest that while SOC can enable spin current in non-magnetic materials such as MoS₂, it would, on the other hand, hinder the generation of pure spin current in some cases. Also, SOC should be treated rigorously when studying both the spin current and the charge current.”

- On page 17, we added, “The predicted BSPV and light-induced pure spin current do not have special requirements except for inversion symmetry breaking, and can be readily integrated with existing semiconductor technologies.”

Below is our point-to-point response to the reviewer’s comments.

1) While it hasn't been called "injection current", the idea of generating spin currents through photoexcitation of semiconductors such as GaAs has been around for a while. This is clear from the fact that Fig 1a of PRL 96, 246601 (2006) has some at least superficial resemblance to Fig 1 of the current manuscript though the role of mirror symmetry is not as emphasized.

Reply: We thank the reviewer for pointing out this interesting work from A. L. Smirl and H. M. van Driel groups. Actually, we have cited a similar but earlier work from the same groups (Ref. 18, [PRL 90, 136603 (2003)]). These works, together with some other relevant works focusing on the charge current, e.g., [PRL 76, 1703 (1996), PRL 78, 306 (1997), etc.], employed the two-color quantum interference control (QUIC). As we will elaborate on below, this mechanism is fundamentally distinct from that in our work, although it is also an optical approach.

1) A prominent difference is that the QUIC requires two beams with frequencies ω and 2ω simultaneously. And the spin/charge current generation is dependent on the relative phases of these two beams (quantum interference). On the other hand, our approach requires only one beam and does not require stringent phase-matching conditions, which is more convenient in practice.

2) QUIC is a (at least) third-order effect. The charge and spin current originate in the interference between the 2ω beam and the ω beam, which is $\propto E(-2\omega)E(\omega)E(\omega)$, where $E(\omega)$ is the Fourier component of the electric field at frequency ω . Besides, the optical excitation from the ω laser beam is a two-photon process and should be a fourth-order effect $\propto E(-\omega)E(+\omega)E(-\omega)E(\omega)$. These facts are evident in two theoretical papers [PRL 76, 1703 (1996), PRL 85, 5432 (2000)]. As a higher-order effect, QUIC would generally be less efficient and smaller in magnitude than the second-order effect in our work.

3) Specific to spin current generation, QUIC requires a narrow frequency window “so that there are no transitions from the (spin) split-off band” [PRL 85, 5432 (2000)]. Such a window is about

0.3 eV in GaAs. On the other hand, our mechanism is applicable for a wide frequency window, as shown in Figures 2-4 in the main text.

On the other hand, QUIC has its own advantages. For example, it does not require inversion symmetry breaking (as it being a third-order nonlinear optical process). Besides, the relative phase between the two beams could provide another degree of freedom to control the spin and charge currents. We agree with the reviewer that the approach for obtaining *pure* spin current in these works is similar to that in our work – once we have counter-propagating electrons with opposite (or at least different) spin polarization, then a pure spin current arises. Actually, this approach is also used in e.g., spin Hall effect [*PRL* **95**, 226801 (2005)]. However, for spin current generation QUIC is a distinct mechanism from the bulk spin photovoltaic effect proposed in our work, as we described above.

2) As I understand it from the manuscript, the mirror symmetry leads to "a pure spin current can be realized, which is highly desirable for energy-efficient spintronics." This confuses me because as pointed out in Ref 1, a pure spin current does not guarantee lack of dissipation. Specifically, in this case, the optical excitation generates electron-hole pairs, which are highly energetically excited. I expect the relaxation of this excitation to lead to significant dissipation at some point.

Reply: We thank the reviewer for pointing out this misleading statement. We did not intend to claim that the generation of spin current, or the flow of spin current, is free of energy dissipation. We intended to express that the pure spin current may be more favorable than a non-pure spin current (mixed with charge currents) for spintronic applications. This is because an accompanying charge current may lead to undesired side effects such as charge accumulation and additional dissipations. We have removed the “energy-efficient” and revised this statement as

“a pure spin current can be realized, which does not carry charge degree of freedom and is more favorable than a non-pure spin current for many spintronics applications.”

We have also revised statements about energy efficiency in several other places in the main text to avoid similar misunderstandings.

Next, we show that for spin current generation, the temperature rise due to the energy dissipation is generally not significant. Here we take monolayer MoS₂ as an example. For the bulk spin photovoltaic effect studied in this work, the main energy consumption is the photon absorption due to interband transitions (electron-hole pair generation), and the absorbance is $A = 1 - \exp\left[-\frac{\omega}{c\epsilon_0}\epsilon^i(\omega)d\right] \approx \frac{\sigma^r(\omega)}{c\epsilon_0}d$, where ϵ_0 is the vacuum permittivity, ϵ^i is the imaginary part of the dielectric function, σ^r is the real part of the optical conductivity (one has $\epsilon(\omega) = \epsilon_0 + \frac{i\sigma(\omega)}{\omega}$). d is the thickness of the material, which is taken as 0.6 nm for MoS₂. The energy consumption rate per unit area is

$$\begin{aligned} P &= AI & (R1) \\ &= \frac{\sigma^r(\omega)}{c\epsilon_0}d \cdot \frac{\epsilon_0 c}{2}E^2 \\ &= \frac{\sigma^r(\omega)d}{2}E^2 \end{aligned}$$

where $I = \frac{\epsilon_0 c}{2}E^2$ is the intensity of the light. From our *ab initio* calculations, at $\omega = 3$ eV one has $\sigma^r(\omega) = 4 \times 10^5 \Omega/\text{m}$. In the main text, we showed that light with electric field strength on the order of $E = 100$ V/m would be able to generate a detectable spin current. With this field strength, the energy consumption power is only $P = 1.2 \times 10^{-4}$ W/cm², which is rather small. Here we calculate the temperature rise under an electric field of $E = 1$ MV/m, which is much stronger, but readily available with laser technology. Under this field strength, one has $P = 1.2 \times 10^4$ W/cm². Assume that MoS₂ is put on a substrate with thermal conductivity κ and thickness l_{subs} . If a continuous wave (CW) light is used, then the steady-state temperature rise can be roughly estimated from

$$\Delta T_{\text{CW}} = \frac{P}{\kappa}l_{\text{subs}} \quad (R2)$$

Assuming that $l_{\text{subs}} = 1 \mu\text{m}$ and $\kappa = 100 \text{ W} \cdot \text{m}^{-1} \cdot \text{K}^{-1}$, then $\Delta T_{\text{CW}} \approx 1.2$ K, which is not significant. On the other hand, if a pulse laser is used, then the temperature rise can be estimated from

$$\Delta T_{\text{pulsed}} = \frac{\tau_{\text{pulse}}PS}{k_B} \quad (R3)$$

Where S is the area of a unit-cell, k_B is the Boltzmann constant, while τ_{pulse} is the duration of the pulse, and is taken as 1 ps here. One can find that $\Delta T_{\text{pulsed}} \approx 0.7$ K, which is not significant as well.

We would like to note that, not all energies absorbed by the materials go to the phonon (lattice) system (non-radiative recombination). They could recombine and re-emit photons. Therefore, the temperature rise in the ion system may be even lower than ΔT_{CW} and ΔT_{pulsed} estimated above.

We have added the discussions above in the Supplementary Materials.

3) It seems that the authors are pointing out that TCIs such as SnSe, SnTe have an advantage relative to the GaAs example because of the double mirror symmetry, which should be emphasized more.

Reply: We thank the reviewer for this helpful suggestion. We have emphasized more the unique role of double mirror symmetry for generating pure spin current. We have also emphasized more on the inversion-spin rotation symmetry \mathcal{PS} .

On page 4, we added “Particularly, when double mirror symmetries, or inversion-spin rotation symmetry \mathcal{PS} are present, the charge current would all vanish (even in the transverse directions), while the spin current can survive. These results are useful not only for generating pure spin currents, but also for material characterization.”

On page 15, we added “There may be other systems that possess double mirror symmetries, such as monolayer FeSe⁷³. They may also be good candidates for pure spin current generation.”

We have also explicitly discussed the role of double mirror symmetries in several other places in the manuscript, particularly in the *Surface States of Topological Materials* section.

4) A more serious omission in terms of references in my view is PHYSICAL REVIEW B 102, 081402(R) (2020), which has proposed an injection-based pure spin photocurrent in the quantum spin-Hall edge states of TMDs. The mechanism is quite similar to the one in the present work, though this is for a one-dimensional edge in a two-dimensional system.

Reply: We thank the reviewer for pointing out this paper [*PRB* **102**, 081402(R) (2020)]. This work uses patterned graphene nanoribbons with antiferromagnetic (AFM) edge states. Actually, we were aware of the paper when preparing our manuscript. But we decided not to cite this paper because:

1) First, it requires exquisite designing and fabricating processes. As indicated in the paper, the graphene nanoribbon needs to be patterned with triangle anti-dots, and it needs to “have opposite band structures and anti-symmetrical spin density for the two leads in their ground state”. These conditions are stringent. For example, the edge states are not necessarily AFM if a different patterning structure is used.

2) Secondly, a more serious problem is that the “pure spin current” “with spatial inversion symmetry” claimed in this paper may be *erroneous*. As indicated in this paper, the carbon atoms on the edges of the anti-dots have AFM spin coupling. Therefore, the system actually does *not* have spatial inversion symmetry, when the magnetism is taken into consideration. Hence, both the nonlinear spin current and the nonlinear charge current are allowed, according to our symmetry analysis. This is vividly illustrated with AFM bilayer MnBi_2Te_4 in our work. The atomic structure of AFM bilayer MnBi_2Te_4 also has inversion symmetry, but the anti-ferromagnetic moments on the upper and lower layers break the inversion symmetry. As shown in Figure 3 in our manuscript, if SOC is not taken into consideration, then the total charge current would be zero, consistent with the claim in [*PRB* **102**, 081402(R) (2020)]. However, if we consider SOC, which transfers the inversion asymmetry in spin degree of freedom to the orbital degree of freedom, then the charge current would be non-zero as well. In [*PRB* **102**, 081402(R) (2020)], the “pure spin current” comes from the fact that SOC is ignored, and that the spin up and down states are treated separately. If SOC is rigorously considered, then the charge current should appear.

On the other hand, in the present work, we clarify that mirror symmetry can lead to pure spin current (with possible charge current in the transverse direction). Also, in some cases when double mirror symmetries exist, the charge current can be totally forbidden (absent even in the transverse directions). Besides, we propose that inversion-spin rotation symmetry \mathcal{PS} can guarantee pure spin current without any charge current as well, which may be realized in e.g.,

skyrmion systems, or magnetic materials with canted or all-in-all-out magnetic configurations, etc. This is based on careful symmetry analysis and is robust even under strong SOC effects.

Finally, we would like to remark that [PRB 102, 081402(R) (2020)] is not dealing with “the quantum spin-Hall edge states of TMDs”, but edge states of graphene nanoribbons. However, we could not find a work that studies “injection-based pure spin photocurrent in the quantum spin-Hall edge states of TMDs”. Thus, we focused on [PRB 102, 081402(R) (2020)]. Two other works [PRB 100, 195410 (2019)] and [PRL 115, 166804 (2015)] study the spin currents in 2H TMDs (which are not quantum spin Hall insulators). But they are also different from our work. Specifically, [PRB 100, 195410 (2019)] requires the proximity effect with magnetic material, and the authors did not claim “pure” spin current (actually it should not be pure spin current). On the other hand, [PRL 115, 166804 (2015)] requires the 2H-WSe₂/2H-MoSe₂ heterostructure and two spatially varying laser beams applied simultaneously.

5) Apart from motivational issues that I raised above, I have a few suggestions for the presentation of the formalism. The symmetry argument presented between lines 125-135, which is one of the main points of the manuscript seems to be a bit sloppy. This is because while Eq. 2, whose numerator is being discussed depends on off-diagonal matrix elements of the spin and velocity operators, they appear to be treated here as numbers. Also, it would be good to clarify this.

Reply: We thank the reviewer for these helpful comments. We have cleared up the confusions and explicitly added the matrix indices, and revised these arguments as

“Next, we consider symmetry constraints on the conductivity tensor. First, we note that the numerators are composed of terms with format $N_{mnl}^{iabc} = j_{mn}^{a,s^i} v_{nl}^b v_{lm}^c$ with $i \neq 0$ for spin current and use $N_{mnl}^{0abc} = v_{mn}^a v_{nl}^b v_{lm}^c$ for charge current. Under spatial inversion operation \mathcal{P} , one has $\mathcal{P}v_{mn}^a(\mathbf{k}) = -v_{mn}^a(-\mathbf{k})$, $\mathcal{P}s_{mn}^i(\mathbf{k}) = s_{mn}^i(-\mathbf{k})$, and $\mathcal{P}j_{mn}^{a,s^i}(\mathbf{k}) = -j_{mn}^{a,s^i}(\mathbf{k})$. Thus $\mathcal{P}N_{mnl}^{iabc}(\mathbf{k}) = -N_{mnl}^{iabc}(-\mathbf{k})$. On the other hand, the denominator is invariant under \mathcal{P} , thus all components (including charge and spin) of σ_{bc}^{a,s^i} should vanish after a summation over $\pm\mathbf{k}$ in \mathcal{P} -conserved systems. Therefore, the inversion symmetry \mathcal{P} has to be broken to give nonvanishing σ_{bc}^{a,s^i} . Regarding time-reversal operation \mathcal{T} , one has $\mathcal{T}v_{mn}^a(\mathbf{k}) = -v_{mn}^{a*}(-\mathbf{k})$, $\mathcal{T}s_{mn}^i(\mathbf{k}) = -s_{mn}^{i*}(-\mathbf{k})$

(Here \cdot^* indicates complex conjugate of \cdot). For charge current, one has $\mathcal{T}N_{mnl}^{0abc}(\mathbf{k}) = -N_{mnl}^{0abc*}(-\mathbf{k})$. Thus, the real and imaginary part of N_{mnl}^{0abc} are odd and even under \mathcal{T} , respectively. The imaginary part of $N^{0abc}(\mathbf{k})$ contributes to the total charge conductivity after the summation over $\pm\mathbf{k}$ in a \mathcal{T} -conserved system. Similarly, for spin- i current ($i \neq 0$), one has $\mathcal{T}N_{mnl}^{iabc}(\mathbf{k}) = N_{mnl}^{iabc*}(-\mathbf{k})$, thus it is the real part of $N^{iabc}(\mathbf{k})$ that contributes to the total spin conductivity.”

We have also added the matrix indices in Table I and several other places in the main text.

6) Another technical aspect is the treatment/interpretation of the scattering time τ , which plays a somewhat important role. In the shift mechanism the time τ is a dephasing or scattering time for the electron-hole pair rather than the equilibration time suggested here. This has been recently clarified in PHYSICAL REVIEW B 101, 045201 (2020), where the second order response formula for the charge BVPE is shown to be equivalent to the dipole moment of electron hole pairs if τ is interpreted as a scattering time. In this sense, the injection mechanism is rather physically different and generates a current from a difference of velocities of the different excited charge carriers. This current survives for as long as the electrons/holes maintain their direction, leading to a different interpretation for τ . I think these subtleties are lost in the second order response formula. Also the electron-hole pair interpretation provides a direct understanding of the symmetries in table 2. Shift currents occur when the excited pair has a dipole moment. Injection currents on the other hand are a result of unequal excitation of carriers of different velocities.

Reply: We thank the reviewer for these insightful comments. We agree that the scattering time τ is rather important. Here we would like to use the charge current as an example to illustrate the role of τ , which is more straightforward, while a similar analysis applies to spin current.

The photocurrent is $j^a = \sigma_{bc}^a E^b E^c$. First, we note that a charge current j is odd under time-reversal symmetry \mathcal{T} , while electric field E is even under \mathcal{T} . If the system is non-magnetic, and we use linearly polarized light (LPL), then it seems that \mathcal{T} should be preserved. In this case, it seems that σ_{bc}^a should be zero, because the j^a is odd under \mathcal{T} , while $E^b E^c$ is even. However, in practice, the nonlinear photocurrent does exist, which is the shift current. Actually, \mathcal{T} is broken

by dissipation, which is characterized by τ . Therefore, the dissipation τ is indispensable for the shift current, although the shift current conductivity σ_{bc}^a is independent of τ .

This point can also be verified mathematically. The nonlinear photoconductivity is,

$$\sigma_{bc}^a(0; \omega, -\omega) = -\frac{e^2}{\hbar^2 \omega^2} \int \frac{d\mathbf{k}}{(2\pi)^3} \sum_{mnl} \frac{f_{lm} v_{lm}^b}{\omega_{ml} - \omega + i/\tau} \left(\frac{v_{mn}^a v_{nl}^c}{\omega_{mn} + i/\tau} - \frac{v_{mn}^c v_{ml}^a}{\omega_{nl} + i/\tau} \right) \quad (\text{R4})$$

Under time-reversal \mathcal{T} operation, one has $\mathcal{T}v_{mn}(\mathbf{k}) = -v_{mn}^*(-\mathbf{k})$, where $*$ indicates the complex conjugate. Thus, the numerator in Eq. (R4), $N_{mnl} = v_{mn}v_{nl}v_{lm}$, would behave as $\mathcal{T}N_{mnl}(\mathbf{k}) = -N_{mnl}^*(-\mathbf{k})$, while the denominator is invariant under \mathcal{T} . After the summation over $\pm\mathbf{k}$, the numerator becomes purely imaginary. No dissipation indicates $\tau = \infty$ and $\frac{i}{\tau} = 0$. In this case, the denominator would be purely real. Therefore, under LPL the whole formula is purely imaginary, and cannot contribute a static current, which should be a real number. Therefore, from a mathematical point of view, a finite τ is indispensable.

These arguments agree with those in [PRB **101**, 045201 (2020)]. That is, τ describes some processes that lead to dissipation, and thus 1) absorb energy from light and 2) break time-reversal symmetry. In [PRB **101**, 045201 (2020)] τ was suggested to be the scattering time with phonon. But in our view, it could also be the scattering with impurities, etc. Phenomenologically, one should have $\frac{1}{\tau} = \frac{1}{\tau_{\text{phonon}}} + \frac{1}{\tau_{\text{impurities}}} + \dots$; that is, τ incorporate contributions from all sources of dissipations. For convenience, we adopted the constant relaxation time approximation and used a constant τ for all modes (band index n and wavevector k). But in reality, τ should be mode-dependent (e.g., different for electrons and holes) and incorporates the subtleties described above. More discussions on τ can be found in our reply to the 3rd comment of Reviewer #2 (on page 16-17 of this document).

We have added the discussions about the role of τ on page 7 of the main text,

“We would like to briefly discuss the carrier lifetime τ . It has a rather important role. Here we use the charge current as an example to illustrate the role of τ , which is more straightforward. Similar analysis applies to spin current. The photocurrent is $j^a = \sigma_{bc}^a E^b E^c$. First, we note that a charge current j is odd under time-reversal symmetry \mathcal{T} , while electric field component E is even under \mathcal{T} . If the system is non-magnetic, and we use linearly polarized light (LPL), then it seems

that \mathcal{T} should be preserved. In this case, it seems that σ_{bc}^a should be zero, because the j^a is odd under \mathcal{T} , while $E^b E^c$ is even. However, in practice, the nonlinear photocurrent does exist, which is the shift current. Actually, \mathcal{T} is broken by dissipation, which is characterized by τ . Therefore, the dissipation τ is indispensable for the shift current, although the shift current conductivity σ_{bc}^a is independent of τ .”

Regarding the physical mechanisms of shift and injection currents, they are more evident when we use the length gauge. In the Supplementary Materials, we showed that the velocity gauge and the length gauge are equivalent. In the length gauge, one has (Eqs. (S27, S28) in the SM, Section 2),

$$\begin{aligned}\zeta_{aa}^c(0; \omega, -\omega) &= -\frac{i\pi e^3}{2\hbar^2} \int \frac{d\mathbf{k}}{(2\pi)^3} \sum_{n,m} f_{nm} R_{nm;c}^a |r_{mn}^a|^2 \delta(\omega_{mn} - \omega) \\ \eta_{ab}^c(0; \omega, -\omega) &= \frac{\pi\tau e^3}{2\hbar^2} \int \frac{d\mathbf{k}}{(2\pi)^3} \sum_{n,m} f_{nm} \Delta_{mn}^c [r_{mn}^a, r_{nm}^b] \delta(\omega_{mn} - \omega)\end{aligned}\quad (\text{R5})$$

Here ζ_{aa}^c and η_{ab}^c are the shift and injection current conductivity, respectively. $|r_{mn}^a|^2$ and $[r_{mn}^a, r_{nm}^b]$ are proportional to the interband transition rate (electron-hole pair generation rate) under linearly and circularly polarized light, respectively. $R_{mn;c}^a = \frac{\partial \phi_{nm}}{\partial k^a} + \xi_{nn}^a - \xi_{mm}^a$ can be regarded as the dipole moment of the electron-hole pair, with ξ_{nn}^a as the center of the lattice-periodic wavefunction of band n (for details see SM). $\Delta_{mn}^c = v_{mm}^c - v_{nn}^c$ is the velocity difference between the conduction and valence bands. These length gauge formulae are reminiscent of the Fermi’s golden rule. The physical meanings of the shift and injection currents are thus evident, which agrees well with the interpretations suggested by the reviewer. Hence, we also added these discussions in the last paragraph on page 15 of the main text:

“The shift current mechanism comes from the fact that the wavefunction center of the electrons and holes are different, leading to an electric dipole upon electron-hole pair generation. On the other hand, the injection mechanism comes from the fact that the electrons and holes have different velocities, leading to a net current. These facts are more evident if we transform Eq. (2) into the length gauge, as shown in the SM.”

The detailed discussions above have been added in the Supplementary Materials.

In summary, while I think the work is technically interesting, I am not sure I am convinced that the authors have clarified the motivation for this work that would lead to a high level of impact suggested for publication in Nature Communications.

Reply: We thank the reviewer for these comments. We hope that we have appropriately addressed the concerns of the reviewer. We believe that our work is of broad impact for publication in Nature Communications.

Reviewer #2:

This is a timely work which shows theoretically how the nonlinear optical effect could be used to generate pure spin photocurrent in non-centrosymmetric crystals. These findings are illustrated on examples from three classes of materials, transition metal dichalcogenides (MoS₂), antiferromagnetic topological insulator (MnBi₂Te₄), and the surface state of the topological crystalline insulator (SnTe). Both the topic of generating (electrically, or optically) spin current and the considered materials classes are actively studied, so the findings of this manuscript have potential many implications. Specifically, as noted, for example, in Ref. 1, spin current provides an important element for spintronic applications as well as a tool to probe materials properties. Methods which would allow for a robust generation of pure spin current therefore have broad ramifications. The authors complement well their symmetry analysis and conductivity calculations by the materials-specific ab-initio calculations.

Reply: We appreciate these positive and encouraging comments from the reviewer.

To provide a better context of this work and further explain its relevance, it would be helpful to give several explanations and clarify its underlying assumptions.

1) Adding a spin degree of freedom can generalize various photovoltaic effects that have been studied in common semiconductors and their junctions. See, for example, Appl. Phys. Lett. 79, 1558 (2001), Phys. Rev. Lett. 88, 066603 (2002), or related experiments, Nat. Comm. 4, 2068

(2013), which may be helpful to mention. Since the spin current is not conserved it may also be important to identify the position at which it is evaluated. Even in a simple p-n junction, illuminated by circularly polarized light, a different voltage dependence of the spin and charge currents allows for a generation of pure spin current.

Reply: We thank the reviewer for these helpful comments and references. We have carefully read these interesting works. They are based on mechanisms reminiscent of the p-n junctions used in solar cells. This is distinct from our work, which is realizable in homogeneous materials and does not require hetero-junctions. Also, we do not need circularly polarized light. We have cited these works in the first paragraph on page 2 of the revised manuscript:

“Also, a spin current can be generated with a mechanism reminiscent of the p-n junction in solar cells^{25–27}.”

2) This non-conservation of the spin current has led to lots of debates about its definition, when the prediction of the spin Hall effect from Refs. 10, 11, was revisited 30 years later. It would help to clarify how the employed definition for spin current compares to the one from Phys Rev. Lett. 96, 076604 (2006), which also recalls a cautionary work of Rashba [Phys. Rev. B 68, 241315(R) (2013)].

Reply: We thank the reviewer for these insightful comments. Indeed, the definition of spin current is still under some debate. The conventional definition $\hat{j}_1 = \frac{1}{2}(\hat{v}\hat{s} + \hat{s}\hat{v})$ is indeed not well defined, although it is convenient, physically appealing, and extensively employed in many works until today. When SOC is taken into account, spin component is not a good quantum number, and this spin current is not conserved. Also, as suggested in Rashba’s work [PRB 68, 241315(R) (2003)], this definition would lead to a non-zero spin current even if an inversion asymmetric insulator is in equilibrium (no electric field, light, etc.). There are lots of debates, and there are also works claiming that we do not need to modify this definition [e.g., PRB 77, 035327 (2008)].

The definition in [PRL 96, 076604 (2006)], which is $\hat{j}_2 = \frac{d(\hat{r}\hat{s})}{dt}$, can be conserved in some systems. However, it requires that “spin generation in the bulk is absent due to symmetry

Figure R1 $\alpha_z \equiv \frac{\|[\hat{H}, \hat{s}^z]\|}{\|\hat{H}\| \cdot \|\hat{s}^z\|}$ for (a) MoS₂ and (b) MnBi₂Te₄.

reasons”. In other words, it requires the bulk integration $\frac{1}{V} \int dV \mathfrak{T}(r) = 0$, where $\mathfrak{T}(r)$ is the torque density on the spins. However, this is not true under external light, which is necessary for our work. An intuitive picture is, under a circularly polarized light, the angular momentum of the photons can be transferred into the electron system, which obviously leads to $\frac{1}{V} \int dV \mathfrak{T}(r) \neq 0$. Therefore, the definition of $\hat{j}_2 = \frac{d(\hat{r}\hat{s})}{dt}$ is also not correct if the system is under light illumination. Particularly, under strong light, the non-conservation of $\hat{j}_2 = \frac{d(\hat{r}\hat{s})}{dt}$ might be high. On the other hand, the calculation of the spin current with $\hat{j}_2 = \frac{d(\hat{r}\hat{s})}{dt}$ is rather involved in practice (although this definition looks simple). To the best of our knowledge, this definition has only been applied in simple model systems, and to the linear order responses.

Here we roughly estimate the difference between the spin current defined with $\hat{j}_1 = \frac{1}{2}(\hat{v}\hat{s} + \hat{s}\hat{v})$ and $\hat{j}_2 = \frac{d(\hat{r}\hat{s})}{dt}$. Compared with \hat{j}_1 , \hat{j}_2 has an additional term that comes from the torque on the spins [*PRL* **96**, 076604 (2006), see also *PRL* **97**, 236805 (2006)]. This term is proportional to $[\hat{H}, \hat{s}]$. Therefore, the relative difference $\left| \frac{\hat{j}_2 - \hat{j}_1}{\hat{j}_1} \right|$ can be roughly estimated from $\alpha_i \equiv \frac{\|[\hat{H}, \hat{s}^i]\|}{\|\hat{H}\| \cdot \|\hat{s}^i\|}$,

where $\|\cdot\|$ indicates matrix norm¹. We have thus calculated and plotted α_z in the first Brillouin zone for MoS₂ (Figure R1a) and MnBi₂Te₄ (Figure R1b). One can see that α_z is on the order of 0.1 ~ 0.2. From this point of view, one may deduce that the difference between \hat{j}_1 and \hat{j}_2 is indeed not negligible, but in general cases, it would not qualitatively change the main results.

As described above, \hat{j}_2 is not a perfect definition of spin current in the presence of external light, either. It might be possible to find a better definition of the spin current that is conserved even when the system is under light illumination. But this is beyond the scope of the current work, and we would like to leave this for future work.

In the first paragraph on page 6 of the main text, we added,

“We would like to remark that there are lots of debates on the definition of spin current⁴⁰⁻⁴², see SM for detailed discussions.”

Detailed discussions above, including Figure R1, have been added in the Supplementary Materials.

3) What are the employed assumptions for loss mechanisms of pure spin photocurrents? For example, in a simple situation analyzed in 1) in addition to the spin relaxation, the spin current can be lost through carrier recombination. This spin dynamics and spin-orbit coupling for electrons and holes are inequivalent.

Reply: We thank the reviewer for these insightful comments. The loss of the spin photocurrents should come from the scattering with phonons, etc., which leads to the recombination of electron-hole (e-h) pairs. As also pointed out by reviewer #1, the shift mechanism comes from the fact that the electron-hole pair has non-zero electric dipole p in non-centrosymmetric materials. However, such dipole would be lost when the electron and hole recombine. Assuming an e-h pair generation rate of R , and a scattering/recombination time of τ , then the steady-state polarization is $P = Rp\tau$. The current, which is the polarization generation rate, can be obtained

¹ This can be naively understood in the following way. We have $\hat{j}_2 = \frac{d(\hat{r}\hat{s})}{dt} = \frac{d\hat{r}}{dt}\hat{s} + \hat{r}\frac{d\hat{s}}{dt} = \frac{1}{2}\{[\hat{H}, \hat{r}], \hat{s}\} + \frac{1}{2}\{\hat{r}, [\hat{H}, \hat{s}]\}$, where $\{a, b\} = ab + ba$ ensures hermiticity. The first term is just \hat{j}_1 , while the second term comes from the torque on the spins. The ratio between these two terms is (very roughly) $\frac{\|[\hat{H}, \hat{s}]\|}{\|\hat{H}\| \cdot \|\hat{s}\|}$. Rigorously speaking, the position operator \hat{r} needs extra care in infinite solid-state systems.

from $j \propto \frac{P}{\tau} = Rp$, and is independent of τ . On the other hand, the injection mechanism comes from the different velocities of electrons and holes, which is $\Delta_{eh} = v_{ee} - v_{hh}$. The velocity difference leads to a current of $j \propto \rho \Delta_{eh}$, where $\rho = \tau R$ is the steady-state e-h density. Thus, the injection current conductivity is linearly dependent on τ . Some more discussions can be found in our reply to the 6th comment of Reviewer #1 (on page 10-11 of this document).

One can see that both the shift and injection mechanism are dependent on the scattering processes of the electrons and holes. Usually, the scattering time τ is on the order of sub-picoseconds. On the other hand, the spin relaxation time is usually longer, on the order of (sub-)nanoseconds. For example, in [*Nat. Phys.* **11**, 830 (2015)] it was shown that TMD has a spin relaxation time on the order of ns. Similar arguments can also be found in [*Appl. Phys. Lett.* **80**, 1558 (2001), *Nat. Comm.* **4**, 2068 (2013), etc.]. Thus, usually the main loss mechanism should be the scattering of electrons/holes, which leads to the momentum relaxation and the recombination of e-h pairs. Of course, in some cases, the spin relaxation time may be shorter (for example, holes usually have shorter spin relaxation time, and the spin relaxation time may be shorter in the presence of magnetic impurities.). In these cases, we are in a different regime and the spin relaxation would be the main loss mechanism. Approximately, one may expect that $\frac{1}{\tau} = \frac{1}{\tau_{\text{scattering}}} + \frac{1}{\tau_{\text{relaxation}}}$, where $\tau_{\text{scattering}}$ is the e/h scattering time, while $\tau_{\text{relaxation}}$ is the spin relaxation time.

Regarding the difference spin dynamics of electrons and holes, we note that in principle τ should not be a constant, but varies with different band n and wavevector k . The difference between electrons and holes shall be included by using a mode-dependent τ . More rigorously, one should use $\frac{\partial \rho}{\partial t} |_{\text{coll}}$ that includes scatterings, spin dynamics, etc., in the von Neumann equation (Eq. S4 in the Supplementary Materials). One has

$$\frac{\partial \rho}{\partial t} = -\frac{i}{\hbar} [H, \rho] + \frac{\partial \rho}{\partial t} |_{\text{coll}} \quad (2)$$

However, for convenience, we adopt the constant relaxation time approximation and use $\frac{\partial \rho}{\partial t} |_{\text{coll}} \approx -\frac{\rho - \rho_0}{\tau}$. We would like to remark that including these subtleties would not change the essence of the results in our work.

In the last paragraph on page 7 of the main text, we added,

“The main dissipation mechanism here is the scattering of electrons and holes with e.g., phonons. The scattering time τ is usually on the order of (sub)-picoseconds. In some cases, when the spin relaxation time is short, it can be the main loss mechanism as well. Also, in the presence of scattering potentials (from e.g., impurities), there could be skew scattering and side jump, which lead to extrinsic spin/charge currents, as compared with the intrinsic currents discussed in this work, which originates in the intrinsic band structure. Another point we would like to mention is that, here we adopt the constant relaxation time approximation and use a constant τ for all modes (band index n and wavevector k). But in reality, τ should be mode-dependent. (see SM for more discussions)”

Detailed discussions above are added in the Supplementary Materials.

4) Perhaps some clarification for the points above could be given by moving the p.15 comment about focusing only on intrinsic currents to an earlier place in the manuscript. Even in such intrinsic regime, how can one understand the corresponding time/length scales for the presence of spin photocurrents?

Reply: We thank the reviewer for these helpful comments. As discussed above, the shift and injection mechanisms rely on the fact that electrons and holes carry different spins and velocities. Such a process would stop once the electrons or holes scatter and recombine. Thus, the relevant time scale here should be the scattering time τ , which is usually on the sub-picosecond scale. Regarding the length scale, the group velocities of electrons and holes are usually on the order of $v = 10^5 \sim 10^6$ m/s. Thus, the length scale here should be $l = v\tau$, which on the order of tens of nanometers. This should be compared with the mechanism based on junctions, as presented in [*Appl. Phys. Lett.* **80**, 1558 (2001), *Nat. Comm.* **4**, 2068 (2013)]. In these works, the electrons and holes move diffusively, and the relevant length scales are relatively larger, on the order of micrometers.

We have moved the discussions on page 15 to page 7 of the revised manuscript, as shown in our reply to the previous question.

5) For an experimental detection of spin photocurrent, could one use spin extraction [predicted in Phys. Rev. Lett. and measured in Nat. Comm. noted above]?

Reply: We thank the reviewer for these comments. In our understanding, spin injection and extraction describe the following processes. A nonmagnetic semiconductor forms a junction with magnetic material. When the carriers flow from the magnetic material to the nonmagnetic semiconductor, they can be spin-polarized, because there are a different number of spin up and down carriers in the magnetic material. This process is dubbed “spin injection” since the spin polarization is kind of “injected” into the semiconductors from the magnetic material. On the other hand, carriers originally in the semiconductor can also flow into the magnetic material. During this process, spin up and down carriers have different probabilities to enter the magnetic materials. This is somewhat similar to the giant magnetoresistance effect. It was also pointed that either the spin up or spin down carriers can have a higher probability to enter the magnetic materials, depending on the actual condition of the junction [*Science* **309**, 2191 (2005), *PRL* **98**, 046602 (2007)]. If the spin up (down) electrons have a higher probability to enter the magnetic materials, then the semiconductor will be left spin down (up), and thus becomes spin-polarized. This process is dubbed “spin extraction” since the spin polarization is kind of “extracted” from the magnetic materials.

We believe that a similar mechanism can be used to detect the spin photocurrent. We can consider a heterojunction using MoS₂ with a ferromagnetic material. When we shine light on MoS₂, a spin current will be generated. The carriers will tunnel into the magnetic material and lead to a current in it. This current would have a different magnitude if the magnetic moment of the magnetic material is parallel or anti-parallel to that of the spin current. We can monitor this effect by switching the magnetic moment with an external magnetic field, or switching the spin polarization of the spin photocurrent by using a light with different polarization (see Figure 2e of the main text). However, it might be improper to call this effect “spin extraction”. Because spin extraction describes a process that the carriers in a semiconductor become spin-polarized *because of* the junction with magnetic materials. But for the process described above, the electrons in the semiconductor are already spin-polarized before entering the magnetic materials.

6) References are not given in a uniform style. Some journal names are abbreviated, some not. Ref. 10 has typos and repeated authors, extra number appear in Ref. 27, 45, 67. Are Refs. 31, 40, 47 complete?

Reply: We thank the reviewer for these careful observations. We have carefully checked all our references, and manually edited the incorrect renderings of the Mendeley plugin.

This manuscript describes an important opportunity to realize spin photocurrent in a wide range of materials. Following these clarifications, it will be easier to assess its suitability for a broad readership and a publication in Nature Communications.

Reply: We thank the reviewer again for these positive and encouraging comments. We hope that we have appropriately addressed the comments of the reviewer.

Reviewers' Comments:

Reviewer #1:

Remarks to the Author:

Dear Editors,

The manuscript titled "Pure spin photocurrent in non-centrosymmetric crystals: bulk spin photovoltaic effect" has been revised by the authors in response to the comments from both referees. With their response, the authors have addressed most of the main concerns of the referees. Specifically, I am now convinced that their conclusion that "nonlinear optical (NLO) effect can be used to generate pure spin currents" "if the system possesses additional mirror symmetry or inversion-mirror symmetry" is novel and interesting enough to be published in Nature Communication.

In reviewing the background, I only noticed one issue in the introduction. The following paper PHYSICAL REVIEW B 95, 224430 (2017) discusses spin currents in inversion broken but time-reversal preserving spin photocurrents at second order in electric field. This appears quite relevant to the current manuscript and should be cited. However, I agree with the authors that pure spin-currents haven't been discussed.

Apart from this change, I would recommend publication in Nature Communication.

Reviewer #2:
Remarks to the Author:

The authors have provided a balanced and detailed response to both reviewers and made related changes in the main text and Supplementary Information. These changes and additional calculations now provide more accurate statements. For example, it was helpful to downplay the low-dissipation aspect of pure spin currents, improve the symmetry arguments, clarify the novelty of the work, or explain possible loss mechanisms for pure spin current.

Even with these changes one can argue (and the authors mention that) that the fully accurate picture of the underlying phenomena is not yet available. In realistic systems with spin-orbit coupling there remain subtleties about the definition of the spin current. This is mentioned in Ref. 41 and the changes with different definition of spin current are further explored in the present work, including new Fig. S1. Additionally, Ref. 41 notes that there are subtleties in spin currents with the choice of boundary conditions, mentioning an example in spin Hall effect [PRB 72, 241303(R), (2005)], which may also pertain to the present work due to nonuniform illumination or spatially-dependent absorption. However, in the revised manuscript what the authors present about the description of spin current already matches the current state-of-the-art and I expect will provide a valuable guidance for a broad readership interested in studies of nonlinear optical effects in a growing number of materials. Therefore, while the knowledge of spin current will continue to evolve, these findings are likely to motivate both emerging applications and probing surface-state properties of topological materials.

With the interest in verifying these predictions, it may help to discuss more when this picture will break down. For example, at the level of a simple estimate as given in Eqs. (R1)- (R3) and using MoS₂ parameters, what is the maximum magnitude of spin current? Will the predicted NLO behavior qualitatively change with an increase in illumination, or the sample will first be destroyed? These estimates can serve as a guidance analogous to gate-controlled effects in spintronics, which are limited by the characteristic breakdown fields and thus limit the generated excess carrier density.

I expect that the implications of these findings to get intriguing effects from light illumination will become relevant in a growing number of materials. I recommend this manuscript for a publication in Nature Communications.

Reply to Reviewers

Reviewer #1:

The manuscript titled "Pure spin photocurrent in non-centrosymmetric crystals: bulk spin photovoltaic effect" has been revised by the authors in response to the comments from both referees. With their response, the authors have addressed most of the main concerns of the referees. Specifically, I am now convinced that their conclusion that "nonlinear optical (NLO) effect can be used to generate pure spin currents" "if the system possesses additional mirror symmetry or inversion-mirror symmetry" is novel and interesting enough to be published in Nature Communication.

Reply: We thank the reviewer for reviewing our paper again and for all these encouraging comments.

In reviewing the background, I only noticed one issue in the introduction. The following paper PHYSICAL REVIEW B 95, 224430 (2017) discusses spin currents in inversion broken but time-reversal preserving spin photocurrents at second order in electric field. This appears quite relevant to the current manuscript and should be cited. However, I agree with the authors that pure spin-currents haven't been discussed.

Reply: We thank the reviewer for pointing out this relevant paper. We have cited this paper in our introduction as Ref. 28,

“Alternatively, a spin current can be generated with a mechanism reminiscent of the p-n junction in solar cells^{23–25}, quantum interference^{26,27}, or the nonlinear Drude current²⁸.”

Apart from this change, I would recommend publication in Nature Communication.

Reply: We thank the reviewer for recommending the publication of our paper.

Reviewer #2:

The authors have provided a balanced and detailed response to both reviewers and made related changes in the main text and Supplementary Information. These changes and additional calculations now provide more accurate statements. For example, it was helpful to downplay the low-dissipation aspect of pure spin currents, improve the symmetry arguments, clarify the novelty of the work, or explain possible loss mechanisms for pure spin current.

Reply: We thank the reviewer for reviewing our paper again and for all these encouraging comments.

Even with these changes one can argue (and the authors mention that) that the fully accurate picture of the underlying phenomena is not yet available. In realistic systems with spin-orbit coupling there remain subtleties about the definition of the spin current. This is mentioned in Ref. 41 and the changes with different definition of spin current are further explored in the present work, including new Fig. S1. Additionally, Ref. 41 notes that there are subtleties in spin currents with the choice of boundary conditions, mentioning an example in spin Hall effect [PRB 72, 241303(R), (2005)], which may also pertain to the present work due to nonuniform illumination or spatially dependent absorption. However, in the revised manuscript what the authors present about the description of spin current already matches the current state-of-the art and I expect will provide a valuable guidance for a broad readership interested in studies of nonlinear optical effects in a growing number of materials. Therefore, while the knowledge of spin current will continue to evolve, these findings are likely to motivate both emerging applications and probing surface-state properties of topological materials.

Reply: We thank the reviewer for these careful and insightful observations. Indeed, the definition of the spin current remains an issue to be explored and is attracting research interest until today. We will study these issues in future works.

With the interest in verifying these predictions, it may help to discuss more when this picture will break down. For example, at the level of a simple estimate as given in Eqs. (R1)- (R3) and using MoS2 parameters, what is the maximum magnitude of spin current? Will the predicted

NLO behavior qualitatively change with an increase in illumination, or the sample will first be destroyed? These estimates can serve as a guidance analogous to gate-controlled effects in spintronics, which are limited by the characteristic breakdown fields and thus limit the generated excess carrier density.

Reply: We thank the reviewer for these insightful comments. With MoS₂ parameters, when the external field strength is $E = 1 \text{ MV/m}$, the spin current density is $10^8 \frac{\text{A}}{\text{m}^2} \frac{\hbar}{2e}$, and the temperature rise in the sample is estimated to be on the order of 1 K, which is not high. If we further increase the electric field strength, then two main issues will arise. 1) On the experimental side, a strong laser may destroy the sample, as pointed by the reviewer. This might happen for electric field strength above 10 MV/m when one uses a continuous wave laser. In this case, the spin current density is around $10^{10} \frac{\text{A}}{\text{m}^2} \frac{\hbar}{2e}$, which is quite high, while the temperature rise can be as high as 100 K. Note that the temperature rise can be mitigated with better thermal management. Also, the sample may survive in an even stronger electric field if a pulsed laser is used. For example, with a femtosecond laser, the electric field can be as high as 100 MV/m, with a temperature rise of 10 K. 2) On the theoretical side, the perturbation theory used in the current work may fail when the electric field is too strong. The external electric field strength should be compared with the intrinsic interaction strength in the materials, which is usually on the order of $1 \frac{\text{V}}{\text{\AA}} = 10^4 \frac{\text{MV}}{\text{m}}$. From this point of view, the perturbation theory may work up to an electric field strength of 100 MV/m. Above this strength, the error from perturbative expansions may not be ignored and non-perturbation theories may be required.

In summary, with a pulsed laser, our theoretical picture may work up to an electric field strength up to 100 MV/m, when the spin current density is on the order of $10^{12} \frac{\text{A}}{\text{m}^2} \frac{\hbar}{2e}$. This is restricted by both experimental (sample damage) and theoretical (validity of perturbation theory) concerns. With a continuous wave laser, one may have to use an electric field strength below 10 MV/m to keep the temperature rise and sample damage manageable. At this field strength, the spin current density can be $10^{10} \frac{\text{A}}{\text{m}^2} \frac{\hbar}{2e}$ with the monolayer MoS₂ parameters.

We have added the discussions above in the Supplementary Materials.

I expect that the implications of these findings to get intriguing effects from light illumination will become relevant in a growing number of materials. I recommend this manuscript for a publication in Nature Communications.

Reply: We thank the reviewer again for these positive comments and the recommendation for the publication of our paper.